# Doublecortin engages the microtubule lattice through a cooperative binding mode involving its C-terminal domain

**Atefeh Rafiei[1], Sofía Cruz Tetlalmatzi[2], Claire H Edrington[2], Linda Lee[3], D Alex Crowder[3], Daniel J Saltzberg[4], Andrej Sali[4,5], Gary Brouhard[2], David C Schriemer[1,3]***

[1]Department of Chemistry, University of Calgary, Calgary, Canada; [2]Department of Biology, McGill University, Montréal, Canada; [3]Department of Biochemistry and Molecular Biology, University of Calgary, Calgary, Canada; [4]Department of Bioengineering and Therapeutic Sciences, University of California, San Francisco, San Francisco, United States; [5]Department of Pharmaceutical Chemistry, California Institute for Quantitative Biosciences, University of California, San Francisco, San Francisco, United States

*For correspondence: dschriem@ucalgary.ca

**Abstract** Doublecortin (DCX) is a microtubule (MT)-associated protein that regulates MT structure and function during neuronal development and mutations in DCX lead to a spectrum of neurological disorders. The structural properties of MT-bound DCX that explain these disorders are incompletely determined. Here, we describe the molecular architecture of the DCX–MT complex through an integrative modeling approach that combines data from X-ray crystallography, cryo-electron microscopy, and a high-fidelity chemical crosslinking method. We demonstrate that DCX interacts with MTs through its N-terminal domain and induces a lattice-dependent self-association involving the C-terminal structured domain and its disordered tail, in a conformation that favors an open, domain-swapped state. The networked state can accommodate multiple different attachment points on the MT lattice, all of which orient the C-terminal tails away from the lattice. As numerous disease mutations cluster in the C-terminus, and regulatory phosphorylations cluster in its tail, our study shows that lattice-driven self-assembly is an important property of DCX.

## Editor's evaluation

In their manuscript, Rafiei et al., investigate the molecular architecture of the microtubule-associated protein, doublecortin-X, by integrating data from chemical cross-linking experiments and in vitro molecular analysis with available crystallographic and cryo-EM structures. They determine the contribution of individual domains of this protein to microtubule-binding and self-association, providing a molecular framework for how this protein binds cooperatively along the microtubule lattice.

## Introduction

The regulation of microtubule (MT) polymerization is central to the maintenance of cell polarity, intracellular transport, and cell division. Aside from the intrinsic dynamics of the MT itself, polymer nucleation, growth, and disassembly are coordinated by many different microtubule-associated proteins (MAPs) to establish and manage the wider cytoskeleton in a cell-type-dependent manner. One significant regulator of MT dynamics in developing neurons is the X-linked *doublecortin* gene. Doublecortin (DCX) is a 40-kDa MAP, essential for neuronal migration in embryonic and postnatal brain development

(*des Portes et al., 1998a*, *Gleeson et al., 1998*; *Francis et al., 1999*). DCX influences MT rigidity and curvature and selects for the canonical 13-protofilament geometry that is characteristic of most eukaryotic MTs (*Moores et al., 2004*; *Jean et al., 2012*; *Bechstedt and Brouhard, 2012*; *Bechstedt et al., 2014*; *Ettinger et al., 2016*). It also increases the nucleation rate and decreases the catastrophe rate (*Moores et al., 2006*), producing a net stabilization of MTs.

Mutations in the *doublecortin* gene result in lissencephaly ('smooth brain') or subcortical band heterotopia, both of which can generate a spectrum of intellectual disabilities and/or epilepsy (*des Portes et al., 1998b*; *Gleeson et al., 1998*; *des Portes et al., 1998a*, *Gleeson et al., 1999*). These mutations impair binding of DCX to MTs (*Taylor et al., 2000*) but the structural basis for this effect remains unclear. DCX consists of a tandem repeat of Doublecortin-like (DC) domains possessing 27% sequence identity, an N-terminal tail (N-tail), a linker region, and a long Serine/ Proline-rich C-terminal tail (C-tail) (*Burger et al., 2016*). Both of the DC domains and the linker region are necessary for the MT stabilization effect (*Taylor et al., 2000*; *Kim et al., 2003*), whereas the C-terminal DC domain (CDC) (*Manka and Moores, 2020*) and the C-tail (*Moores et al., 2004*) help establish the preference for 13-protofilament MTs. The C-tail is also implicated in regulating MT–actin interactions (*Fu et al., 2013*; *Jean et al., 2012*; *Tsukada et al., 2005*). Structures are only available for the isolated N-terminal DC domain (NDC) (*Kim et al., 2003*; *Cierpicki et al., 2006*; *Burger et al., 2016*) and the CDC (*Burger et al., 2016*; *Rufer et al., 2018*). For the most part, these structures share the same globular ubiquitin-like fold, although the CDC may adopt a substantially opened form (*Rufer et al., 2018*). Disease mutations cluster in both of the structured domains, suggesting a functional conservation based upon a shared property, possibly a direct interaction with the MT lattice (*Taylor et al., 2000*).

Cryo-electron microscopy (cryo-EM) investigations of the DCX–MT interaction reveal that one of the two structured domain binds to MTs at the junction of four α/β-tubulin dimers, stabilizing the longitudinal protofilament and its lateral contacts (*Fourniol et al., 2010*; *Liu et al., 2012*). Given the modest resolution of the cryo-EM maps and the structural similarity of DC domains, there is still some uncertainty over the identification of the actual interacting DC domain. NDC was designated as the primary contact at the junction, but evidence also suggests that CDC could occupy this site (*Burger et al., 2016*). The binding mode is clouded further by observations that DCX has a propensity to dimerize in vitro (*Caspi et al., 2000*), although analytical ultracentrifugation showed that isolated DCX is monomeric even at relatively high concentrations (*Moores et al., 2006*). However, CDC may adopt an open conformation, allowing it to dimerize through a 'domain swapping' event in some MT-assisted fashion (*Rufer et al., 2018*). A recent study describes a dynamic interaction model in which DCX interacts with the MT through the CDC domain first, then transitions to an NDC–MT-binding mode in the fully assembled MT lattice (*Manka and Moores, 2020*). The various models presented are incompatible. It is difficult to rationalize how a dimerizing DCX could interact with the MT lattice in a manner that allows for both NDC and CDC binding and explain the density observed in the cryo-EM measurements.

Greater clarity on the primary modes of MT engagement would help address the structural basis for the observed mutational effects, as well as help resolve additional structure–function problems. For example, DCX manifests a higher affinity to MT ends compared to the MT lattice (*Bechstedt et al., 2014*) and appears sensitive to MT lateral curvature (*Bechstedt and Brouhard, 2012*) and longitudinal curvature (*Bechstedt et al., 2014*). These observations are difficult to explain through a simple binding mode, particularly given that binding is strongly cooperative and significantly reduces the rate of DCX–MT dissociation (*Bechstedt and Brouhard, 2012*). Further, understanding how DCX is functionally altered by phosphorylation (*Tanaka et al., 2004*; *Graham et al., 2004*; *Shmueli et al., 2006*) and engages coregulatory proteins like Neurabin II (*Tsukada et al., 2005*) and kinesin-3 motor protein (*Liu et al., 2012*; *Lipka et al., 2016*) requires a complete structure of the full-length protein on the MT lattice. Here, we develop a unifying model of the DCX–MT interaction through an integrative structure-building approach, using data from an improved crosslinking mass spectrometry (XL-MS) method (*Ziemianowicz et al., 2019*, *Rafiei and Schriemer, 2019*), together with available cryo-EM and X-ray crystallographic structures. Our results support a DCX–MT interaction model in which NDC binds to MTs at the junction of the four α/β-tubulin dimers, and induces DCX self-association through its CDC and C-tail domains, creating an extended structure capable of capturing different lateral and longitudinal MT interactions.

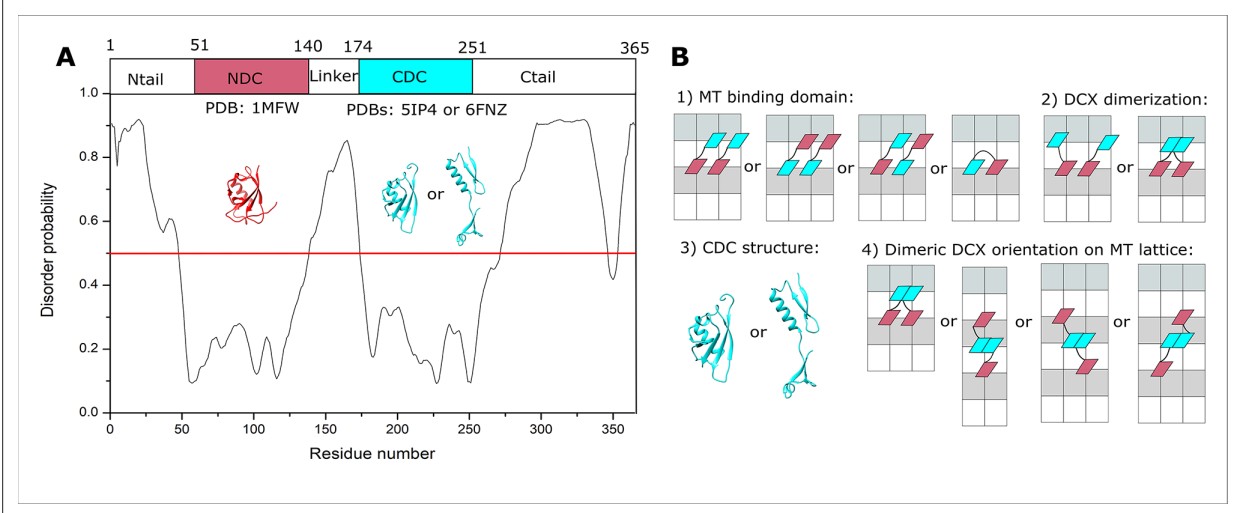

**Figure 1.** Doublecortin (DCX) structure and lattice interaction options. (**A**) The disordered regions of DCX sequence predicted using PrDOS (**Ishida and Kinoshita, 2007**). (**B**) Schematic representations of the four orientational challenges to the elucidation of the DCX–microtubule (MT)-binding mode. In each case, a minimal DCX construct is shown, comprising NDC (red), linker (black), and CDC cyan. α- and β-tubulin are shown with white and light gray rectangles, respectively.

The online version of this article includes the following figure supplement(s) for figure 1:

**Figure supplement 1.** The evaluation of purity and activity of in house purified Doublecortin.

## Results

### Preparation of the DCX–MT construct

DCX possesses two structured domains, each flanked by regions of disorder, which could allow it to adopt various mechanisms of MT lattice engagement (**Figure 1A, B**). The activity of the recombinantly purified DCX was confirmed using three different methods (**Figure 1—figure supplement 1**). In a turbidity assay, adding DCX to purified α/β tubulin-induced MT polymerization, consistent with previous reports (**Taylor et al., 2000**; **Horesh et al., 1999**; **Moores et al., 2006**; **Bechstedt and Brouhard, 2012**). The effect saturates at 10–20 µM DCX, in line with previous claims of a 1:1 binding ratio (DCX:α/β-tubulin) (**Moores et al., 2006**). A pelleting assay confirmed a 1:1 stoichiometry. Finally, using a fluorescence image analysis, we observed MT lengths decrease with increasing DCX concentration, confirming a role in MT nucleation (**Moores et al., 2004**). There were no signs of extensive MT bundling.

### Sampling the equilibrated interaction with photoactivated crosslinking

Many different crosslinking reagents are available for measuring site-to-site distances, but most are not appropriate for structural characterization of dynamic systems. The inherent flexibility of DCX renders it susceptible to 'kinetic trapping' on the MT lattice when using conventional long-lived reagents (**Ziemianowicz et al., 2019**), thus potentially scrambling the sites of interaction. That is, conformations not representative of the structural ensemble can be selected based simply on higher reaction rates. Therefore, we used a heterobifunctional crosslinker (LC-SDA) for MT interactions that has been demonstrated to minimize this effect (**Ziemianowicz et al., 2019**, **Rafiei and Schriemer, 2019**). The first coupling is to accessible nucleophiles through a conventional NHS ester and the second coupling to any surface-accessible residue through laser-initiated carbene chemistry. When applied to the saturated DCX–MT state, the method generated a dense set of 362 unique crosslinks, well distributed among the domains and subunits (**Figure 2A** and **Supplementary file 1**), including 124 interprotein crosslinks between DCX and α/β-tubulin.

These datasets were then used for integrative structure determination. Given the multiplicity of binding modes that are possible between DCX and the MT lattice, we reasoned that an incremental approach to modeling focused on determining the major interaction modes would be necessary for computational efficiency. Thus, we formulated the modeling exercise into four stages: (1) identify the

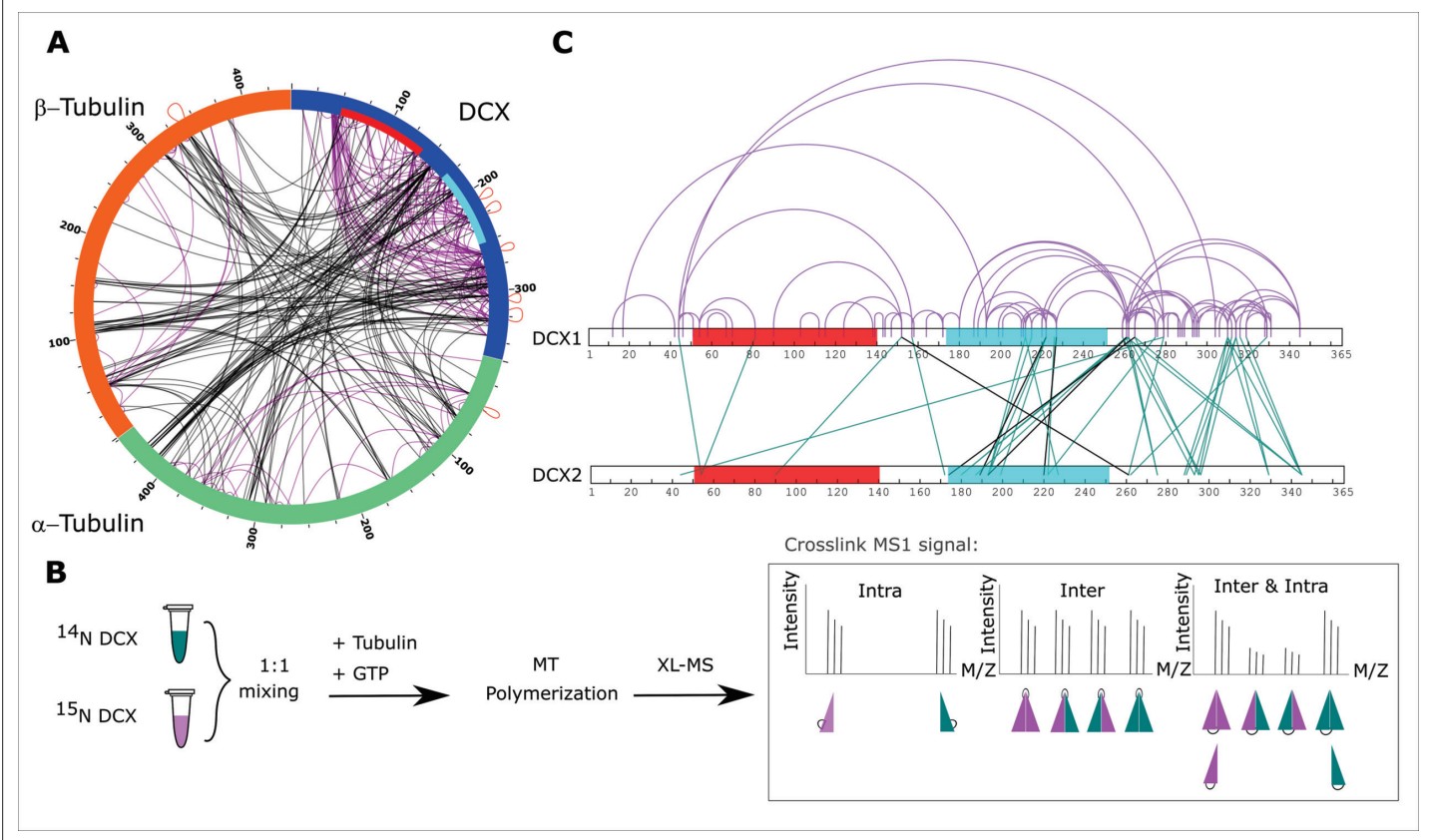

**Figure 2.** Crosslinking mass spectrometry analysis of Doublecortin (DCX). (**A**) Two-dimensional crosslinking map linking α–β-tubulin and DCX at specific residues. NDC and C-terminal DC (CDC) domains of DCX sequence are shown with red and cyan, respectively. Intraprotein crosslinking sites are shown in black and interprotein crosslinks in purple. A subset of interprotein crosslinking sites observed between peptides with a shared sequence are shown in red loops. The crosslinking map is produced using xVis (**Grimm et al., 2015**). (**B**) Inter- and intra-DCX crosslinking sites differentiated using a protein isotopic labeling technique. Three categories are defined, where the ratios of intensities for the inner and outer doublet in the mixed state are variable. The crosslinking map is produced using xiNET online tool (**Combe et al., 2015**). (**C**) Two-dimensional crosslinking map linking DCXs at specific residues. NDC and CDC domains of DCX are shown in red and cyan, respectively. Shared inter and intra XLs are shown as inter (green) and intra (purple) DCX crosslinks. Unique inter-DCX crosslinking sites are shown in black. In all cases, when there is ambiguity in the crosslinking site, a single crosslinking site pair was used for visualization.

The online version of this article includes the following figure supplement(s) for figure 2:

**Figure supplement 1.** $^{15}$N incorporation in purified heavy labeled DCX was assessed by LC-MS analysis.

**Figure supplement 2.** Monitoring the ratio of heterogenous species (Heavy-Light:HL and Light-Heavy:LH) to the homogenous species (Light-Light:LL and Heavy-Heavy:HH), for different crosslinked peptides.

primary binding domain using only crosslink restraints between MT and structured DCX domains, (2) evaluate if MT engagement induces DCX dimerization using inter-DCX crosslink restraints, (3) determine if CDC can adopt a domain-swappable (open state) conformation using only DCX crosslinks, and finally (4) determine the orientation(s) of DCX on the MT lattice, constrained by major modes determined in 1–3 and the full set of crosslinking data. These modeling stages were developed with available crystal structures, cryo-EM maps, and the XL-MS restraints as required, using a four-step workflow (**Rout and Sali, 2019**; **Alber et al., 2007**; **Webb et al., 2018**; **Figure 3**).

## NDC is the main MT-binding domain

A minimal MT 'repeat unit' was established for modeling, consisting of two α-tubulin and four β-tubulin subunits. This repeat unit contains all possible lateral and longitudinal tubulin–tubulin interactions found in the main B-lattice state. As DCX is excluded from A-lattice interactions (**Fourniol et al., 2010**; **Manka and Moores, 2018**), this lattice type was not built into our model representation. The NDC and CDC domains were then tested separately for their occupancy of the major binding

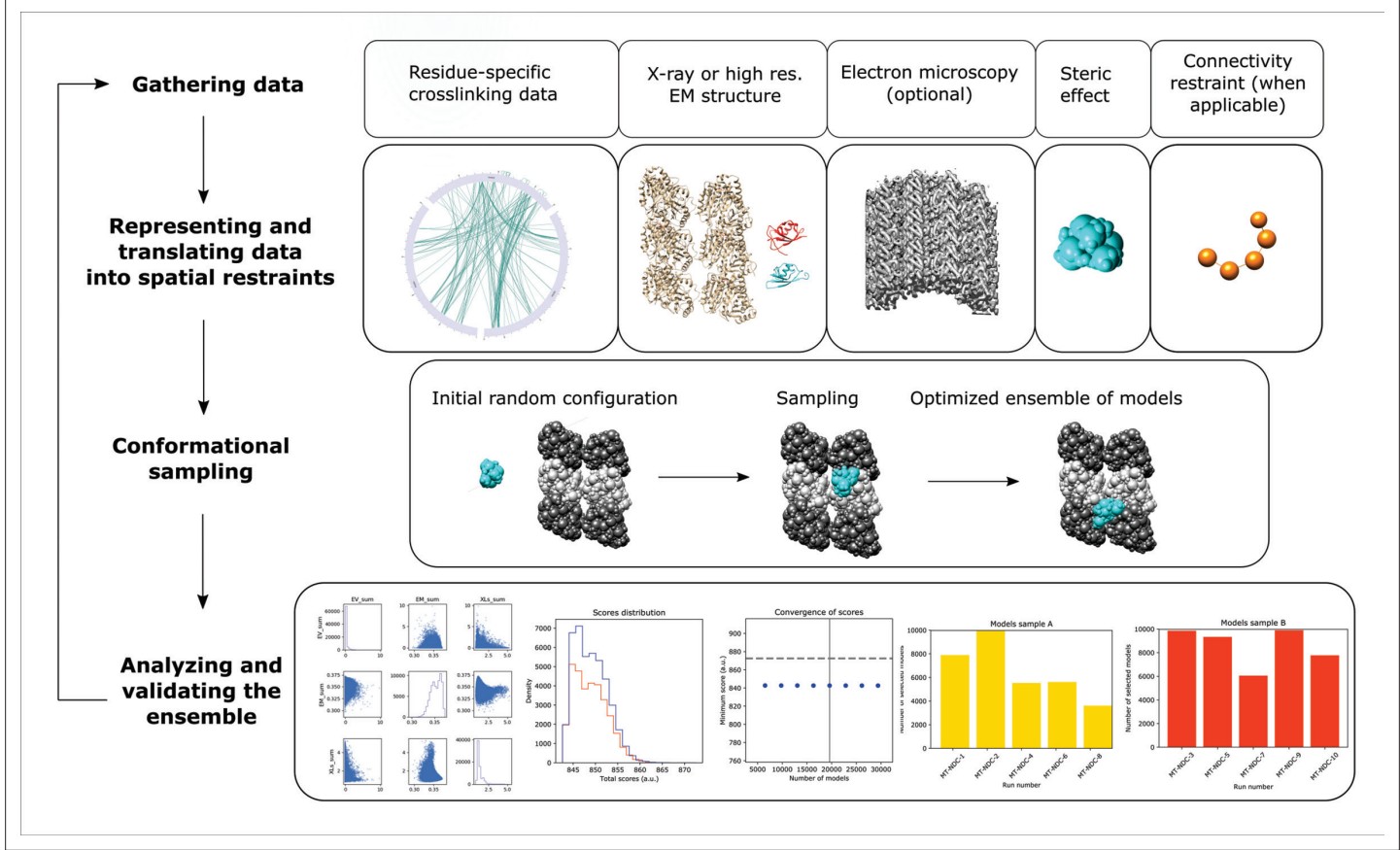

**Figure 3.** Integrative structure determination of microtubule (MT)–Doublecortin (DCX), using four stage workflow. (1) Data gathering, including chemical crosslinking, atomic structures (MT structure, PDB 6EVZ; NDC structure, DCX component of PDB 4ATU; CDC structure, PDB 5IP4), and cryo-EM map EMD 2095. (2) Representation of subunits and translation of the data into spatial restraints, including crosslinking distance restraints, atomic structures, cryo-EM map, and physical restraints (steric effect or excluded volume restraint and connectivy restraint). (3) Monte-Carlo conformational sampling to obtain an ensemble of structures needed to satisfy the input data. (4) Statistical analysis of the computed models and clustering the sample models into distinct groups of structures, followed by the analysis in terms of accuracy and precision.

site. Integrative modeling was performed in two ways: first using only crosslinks, and then crosslinks combined with the available DCX–MT EM density map (*Liu et al., 2012*). This approach allowed us to determine, by comparison, how well the crosslinking data alone could locate the expected binding site and thus validate the quality of the crosslinks. The results are shown in *Figure 4*.

Two major clusters of solutions were obtained for each domain when using the crosslinks alone (*Figure 4—figure supplement 1*, *Figure 4—figure supplement 2*, and *Supplementary file 2*). For NDC–MT, the top cluster contained 48% of all models (cluster precision of 23.4 Å) and identified a site at the junction of the four tubulin dimers resembling the site identified by cryo-EM, albeit with a slightly altered orientation (*Figure 4A, E*). The most accurate models, as measured by RMSD from the expected site also had the highest crosslink utilization rate (*Figure 4A*). The next major cluster, containing 36% of models (cluster precision of 25.0 Å), was identified at the partial binding site at the edge of the repeat unit and thus can be ignored (*Figure 4—figure supplement 3*). The top cluster for CDC–MT contained 50% of the individual models (cluster precision 27.0 Å) and identified a different location, on the interprotofilament junction of two α-tubulin subunits (*Figure 4B* and *Supplementary file 2*). These models are diffusive in their accuracy and crosslink utilization rate. The second cluster, containing 23% of models (cluster precision of 30.0 Å), was identified in the lumenal region of the MT lattice (*Figure 4—figure supplement 3*).

The addition of the EM data to the integrative modeling input, not surprisingly, returned the same binding site for NDC–MT (*Figure 4C*). A single distinct cluster was generated that contained 80% of all models (with a cluster precision 6.0 Å) and it corrected the orientation of the domain (*Figure 4E*).

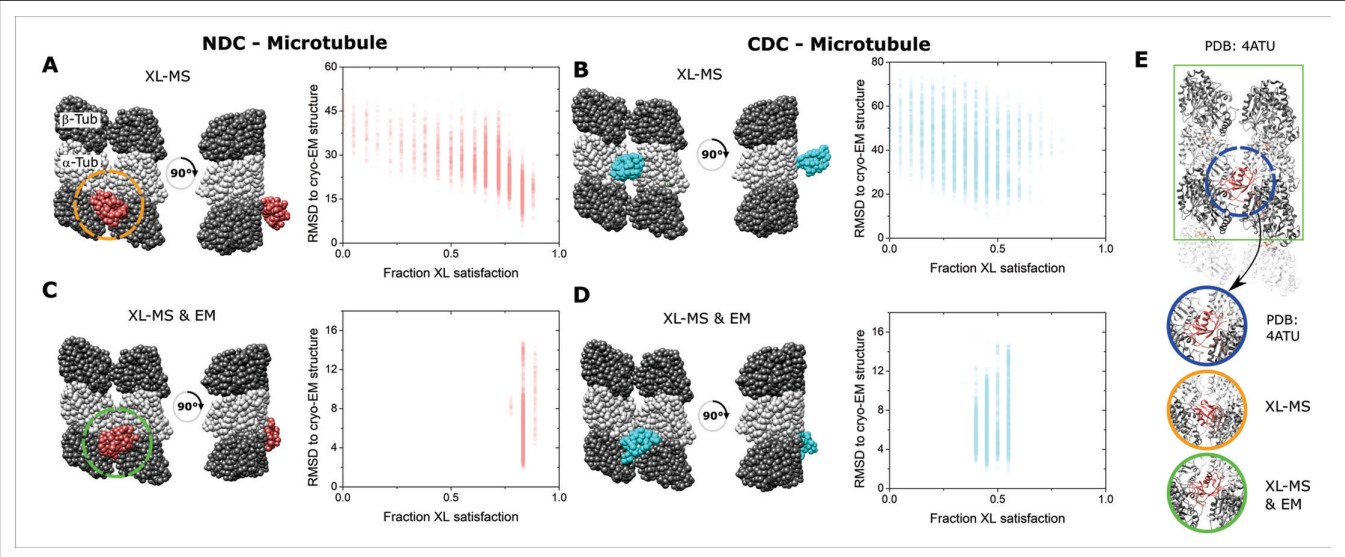

**Figure 4.** Integrative structural modeling for NDC/CDC–microtubule (MT). (**A**) The centroid structure for the main cluster of models produced by IMP for NDC–MT guided exclusively by crosslinking mass spectrometry (XL-MS) restraints. (**B**) The centroid structure for the main cluster of models produced by IMP for CDC–MT guided exclusively by XL-MS restraints. (**C**) The centroid structure for the main cluster of models produced by IMP NDC–MT for XL-MS restraints combined with the EM density map. (**D**) The centroid structure for the main cluster of models produced by IMP CDC–MT guided by XL-MS restraints and combined with the EM density map. The fractional crosslink satisfaction (defined as <35 Å) versus RMSD to the canonical binding site (PDB 4ATU) for all models in the main structural cluster is presented for each modeling scenario. α- and β-tubulin are shown as light and dark gray, respectively. NDC is shown in red, and CDC in cyan. (**E**) Expansions of the NDC orientations from modeling, compared to cryo-EM based NDC–MT structure (PDB 4ATU).

The online version of this article includes the following figure supplement(s) for figure 4:

**Figure supplement 1.** IMP analysis output for MT-NDC.

**Figure supplement 2.** IMP analysis output for MT-CDC.

**Figure supplement 3.** Integrative structural modeling for NDC-MT and CDC-MT (2nd major clusters).

**Figure supplement 4.** Crosslinking-mass spectrometry analysis of DCX-MT using conventional crosslinkers.

Conversely, the addition of the EM data forced the relocation of CDC to the major binding site, well removed from the one generated by crosslinking alone. Although 58% of all models clustered with a precision 5.9 Å, the crosslink utilization rate dropped considerably (*Figure 4D*). Taken together, the high congruency between crosslinking and cryo-EM data confirms the location of NDC at the junction, whereas the variable localization of the CDC and a weaker restraint set suggests, at best, a secondary binding site. Thus, for successive stages of modeling, the NDC was located at the primary binding site and the CDC was left free to move. We note that our modeling used PDB 5IP4 for the CDC domain. A recent CDC structure (PDB 6RF2, *Manka and Moores, 2020*) is structurally very similar (RMSD of 5.1 Å). As this value is lower than the precision of our integrative method, we used 5IP4 throughout.

## DCX self-associates on the MT lattice through CDC and C-tail domains

We next evaluated if DCX could form higher-order assemblies, and if so, which subunits are involved. To differentiate intersubunit crosslinks from intrasubunit crosslinks, we used heavy isotopes ($^{15}$N) installed metabolically during DCX expression. The incorporation of $^{15}$N was assessed by LC–MS/MS, demonstrating >99% incorporation (*Figure 2—figure supplement 1*). Then, a 1:1 mixture of light and heavy labeled DCX ($^{14}$N:$^{15}$N) was used in place of light DCX in sample preparation, followed by cross-linking. Intra- and interprotein crosslinks were differentiated based on the characteristic MS1 pattern of crosslinked peptides. We identified three types of crosslinking signatures reflective of intraprotein crosslinks, interprotein crosslinks, or a mixed state where both types can exist simultaneously although at different levels (*Figure 2B*). This latter category is identifiable through a variable intensity pattern of the 'inner doublet' (*Melchior et al., 2016*). We then explored if the interprotein labeling patterns could be generated without the addition of tubulin, which would indicate some measure of

self-interaction in the free form, possibly dimerization or higher-order associations. We found that only under extreme cases (i.e., DCX denaturation, refolding, and concentrating) could we induce a small amount of interprotein crosslinking, and only at two sites (*Figure 2—figure supplement 2*). Thus, DCX self-association is a MT-dependent phenomenon, consistent with previous reports (*Moores et al., 2006*). In total, we obtained 32 unique intra-DCX crosslink sites, 5 unique inter-DCX crosslink sites as well as 45 unique crosslink sites in the mixed group, both inter-DCX and intra-DCX (*Figure 2C* and *Supplementary file 3*). Interestingly, >80% of the inter-DCX crosslinks identified involve the CDC domain and the C-tail, which indicates that DCX self-associates on the MT lattice through its C-terminal regions. While the inter-DCX crosslinks cannot distinguish between a dimeric state or higher-order assemblies, we chose to proceed with modeling the dimeric state as probable form of self-association based on the crystallographic model (*Rufer et al., 2018*).

## On-lattice dimer shows a preference for an open state

Before attempting to model the full dimeric structure using all available crosslink data, we explored if the subset of DCX–DCX crosslinks could indicate how the DCX–MT interaction might induce a dimerization event. Specifically, we performed integrative modeling to test if the data could distinguish between an interaction dominated by the globular CDC structure observed in the free form (PDB 5IP4, *Burger et al., 2016*) or an open state reflective of a 'domain-swapped' dimerization as suggested by recent crystallographic studies (PDB 6fNZ, *Rufer et al., 2018*). Two full-length DCX molecules were modeled on an expanded MT lattice to allow for all possible orientations of dimerized DCX (i.e., three protofilaments of three α/β-tubulin dimers each, *Figure 5A*). Pairs of NDCs were placed at the confirmed junctional binding sites in four possible orientations: laterally across the protofilaments, longitudinally along the axis of the protofilaments, or in one of two diagonal geometries. These multiple scenarios require assessment as the placement of the anchoring interactions could dictate the success of the dimerization event.

In all modeling scenarios, we obtained major clusters of solutions (>98% of models) with a cluster precision of 21–25 Å (*Figure 5—figure supplement 1* and *Supplementary file 2*), each capable of supporting both globular and domain-swapped modes (*Figure 5A*). An analysis of the overall fit to all DCX–DCX crosslinks shows a weak preference for the latter (*Figure 5B*), but we cannot discriminate with high confidence based on utilization rates alone. However, an inspection of the centroid model for the main cluster of solutions shows a head-to-tail conformation with strong similarity to the domain-swapped structure (*Figure 5C, D*). We note that no symmetry constraints were enforced during the modeling and it only is guided by crosslinking data. Taken together, there appears a preference for the open conformation through a domain swapping event in the CDC. Although dimerization through the globular domains remains possible, a conformational change induced by a lattice interaction could readily explain why dimerization in solution is not possible.

## DCX may not adopt a unique orientation on MTs

We next modeled the DCX–MT interaction with the complete set of crosslinks, to determine if the addition of crosslinks between DCX and MT (in particular) could orient the dimer on the MT lattice. We imposed a set of restrictions based on the findings described above. That is, for the full modeling exercise, we assumed that NDC binds to the MT lattice at the junctional binding site and DCX dimerizes on MT lattice specifically through CDC and C-tail regions. We carried over the degree of ambiguity in the nature of the dimerization event by modeling with both open and globular CDC structures and assessed all four possible orientations for the dimer. For each of the resulting eight modeling exercises, one major cluster was obtained with a sampling precision of 14–20 Å, containing more than 99% of all the individual models (cluster precision of 26–31 Å) (*Figure 5—figure supplement 2* and *Supplementary file 3*). We could detect no preferred orientation for the DCX dimer on the lattice, based on crosslink usage. The distribution of crosslinks shows they accommodate all the orientations equally well (*Figure 6A*). This dispersion suggests that an underlying heterogeneity in lattice engagement is possible.

Finally, to evaluate if the C-terminal tails engage the MT lattice in a preferred orientation and if they participate in stabilizing the interaction, we inspected the density maps of the ensemble of models for each of the orientations (*Figure 6B*). In all cases, regardless of the orientation, the tails are located distal to the lattice and adopt an averaged orientation perpendicular to the long axis of the dimer.

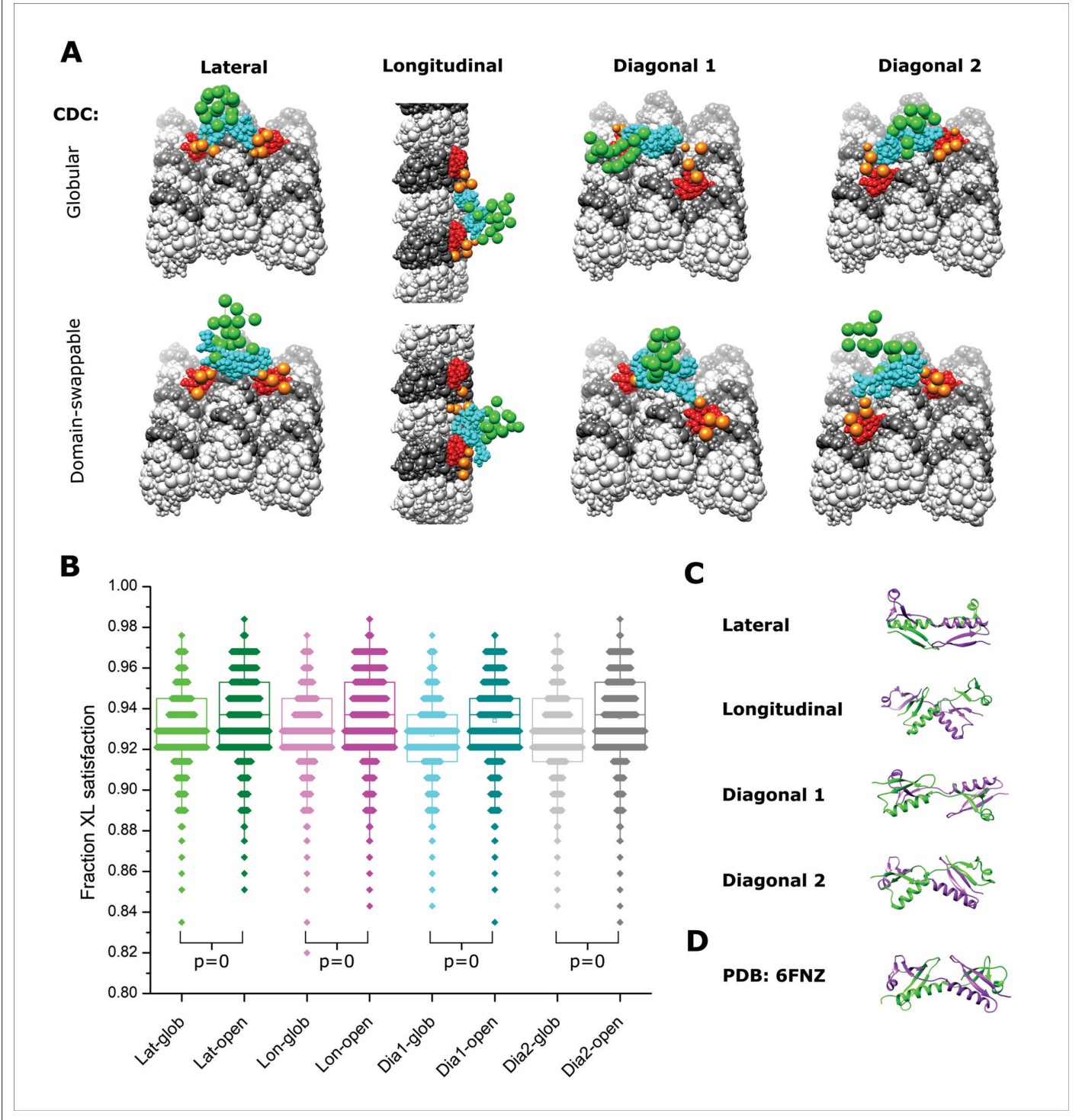

**Figure 5.** Integrative modeling of Doublecortin (DCX) self-assembly on microtubule (MT) lattice. (**A**) The dimeric DCX–MT centroid model of the main clusters of models generated using only DCX–DCX crosslinking restraints. Four different relative positions of fixed NDC on the MT lattice were assessed (lateral, longitudinal, diagonal 1, and diagonal 2); α- and β-tubulin are shown as light and dark gray, respectively. NDC, linker, CDC, and C-tail regions are shown as red, orange, cyan, and green, respectively. CDC structure is represented as either globular or open (domain-swappable) conformations. (**B**) The fractional XL satisfaction (defined as <35 Å) for the main cluster of models generated for each modeling scenario. (**C**) The relative orientation of the dimeric CDC structures in the centroid model of the main cluster using the domain swappable CDC conformation; green is CDC (monomer 1) and purple is CDC (monomer 2). (**D**) The crystal structure of domain-swapped CDC dimer (PDB 6FNZ).

The online version of this article includes the following figure supplement(s) for figure 5:

*Figure 5 continued on next page*

Figure 5 continued

**Figure supplement 1.** IMP analysis output for MT-dimeric DCX using DCX-DCX crosslinking data, where the NDCs were fixed at lateral (A), longitudinal (B), diagonal 1 (C) and diagonal 2 (D) relative positions on the MT lattice.

**Figure supplement 2.** IMP analysis output for MT-dimeric DCX using MT-DCX and DCX-DCX crosslinking data, where the NDCs were fixed at lateral (A), longitudinal (B), diagonal 1 (C) and diagonal 2 (D) relative positions on MT lattice.

## The C-terminal tail plays a minor role in DCX cooperativity

To explore the contribution of the C-terminal tail alone in stabilizing the interaction of DCX on the lattice, we generated an R303X truncation mutation observed in clinical isolates (*des Portes et al., 1998a*), which removes the bulk of this disordered region. We tested whether this mutation reduced the cooperativity of DCX binding to paclitaxel-stabilized MTs (taxol MTs). We mixed a low concentration full-length wild-type DCX-mCherry construct with either a wild type (WT) or a truncated (R303X) DCX-GFP construct (*Figure 7A*). In the case of high cooperativity, we would anticipate soluble DCX to be recruited to the lattice by the DCX already bound. In other words, if there is a high level of WT DCX-mCherry binding, there should also be a high-level DCX-WT-GFP binding. If a C-tail truncation reduces cooperativity, we would expect less binding between DCX-R303X-GFP and DCX-mCherry. This would result in a lower correlation between the mCherry and GFP intensities on the MT lattice.

The interaction of DCX-mCherry and DCX-GFP was imaged and restricted to a mask generated by an image of the taxol MTs (*Figure 7B*). Using a fixed and low concentration of DCX-GFP (0.5 nM), we titrated the DCX-mCherry concentration from 2.5 to 9.5 nM. For each field of view, the correlation between the mCherry and GFP intensities was measured by calculating a Pearson correlation coefficient for the two signals (*Figure 7C*). Our data demonstrate that removal of the C-tail led to a slight reduction in correlation between mCherry and GFP signals, as confirmed by a two-way analysis of variance. In other words, the C-tail contributes slightly to cooperativity between neighboring DCX molecules on the MT.

## Discussion

The function of DCX in MT regulation and brain development is dependent on the full-length protein and we propose that this function is strongly dependent upon its self-association. Our findings support a binding mode that presents DCX primarily as an intralattice dimer, oriented through NDC interactions at the junctional sites, and linked through the CDC domain. Based on the data-directed models that we generated in this study and the high fraction of crosslinking data that was satisfied, a lattice-induced dimerization event is the dominant state at the stoichiometry we explored. However, the association could adopt a variety of orientations on the stabilized MT lattice, consistent with an initial contact complex that involves just a monomer. That is, the flexibility of the interdomain region could support the precession of the monomer, allowing it to capture a suitably positioned additional monomer (*Figure 6C*).

A primary binding mode involving the CDC at the junctional sites is unlikely in the MT polymerization state that was studied here. Although we observed some crosslinks between the CDC and the MT lattice, most of these tend to locate the CDC away from the canonical site and may simply indicate couplings to the dimer as it transiently lays down on the MT lattice. The interaction involving the C-tail domains is particularly intriguing and may indicate that self-association is more extensive than dimerization of the structured domains. An earlier observation highlighted strong binding cooperativity, revealing a Hill coefficient approaching 3 for DCX binding to 13-protofilament MTs (*Bechstedt and Brouhard, 2012*). This remarkable degree of cooperativity suggests that associations beyond dimeric are indeed possible and could involve the long C-tails. At high-density occupation, it is easy to imagine that these tails can associate across dimers to further stabilize the MT lattice (*Figure 6B*). Our observation that the clinical truncation at R303, removing most of the tail, leads to a slight drop in cooperative binding suggests the tail has only a minor role in maintaining associations between DCX molecules on the MT lattice. The tail is not the sole source of cooperativity, as mutations in the structured CDC itself lead to rapid dissociation from the lattice (*Bechstedt and Brouhard, 2012*). Disease specific mutations in both of these regions provide strong evidence for the functional significance of lattice-driven self-association. The binding mode we describe in this study can unify interaction models that appear at first glance to be contradictory. Crystallographic evidence suggests that the

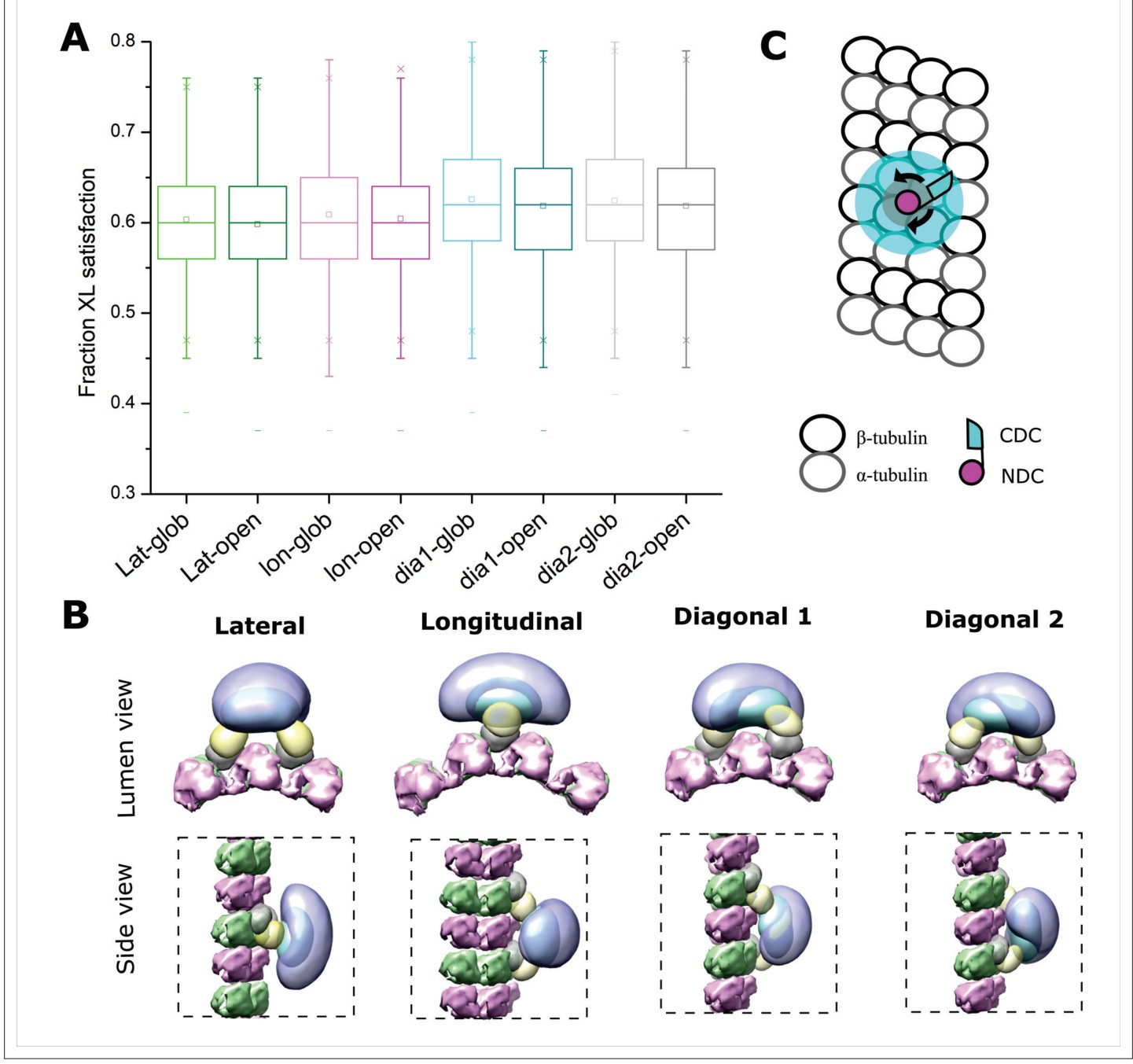

**Figure 6.** Positional evaluation of dimeric Doublecortin (DCX) on microtubule (MT) lattice. (**A**) The fractional XL satisfaction for the main cluster of models generated for dimeric DCX–MT, employing all crosslinking restraints. The four different relative positions of fixed NDC on the MT lattice and the two CDC conformations were assessed. (**B**) The density maps corresponding to the main cluster of the dimeric DCX–MT model, using all crosslinking restraints. The four different relative positions of fixed NDC on the MT lattice were assessed; α- and β-tubulin are shown in pink and green, respectively. NDC, linker, CDC, and C-tail are shown as gray, lemon, cyan, and light purple, respectively. (**C**) The DCX–MT interaction model showing the flexibility of the DCX structure on the MT lattice.

CDC can dimerize under mildly denaturing conditions (*Rufer et al., 2018*) whereas cryo-EM data indicate that both NDC and CDC can bind to the lattice. A recent cryo-EM study by the Moores lab confirms that the CDC domain participates in MT nucleation at the junctional site but as the lattice grows, the sites become mostly occupied with NDC domains (*Manka and Moores, 2020*). The key to resolving this discrepancy is the conformational flexibility of the CDC. We propose a model where

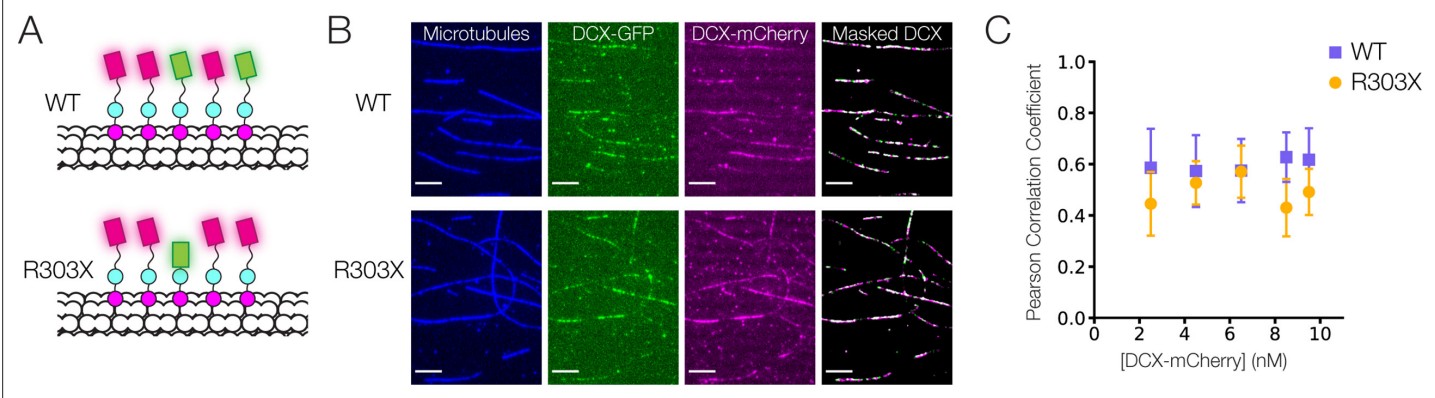

**Figure 7.** Cooperative binding of C-terminal tail truncated Doublecortin (DCX) on microtubule (MT). (**A**) Schematic of DCX-mCherry engaging in cooperative binding with DCX-GFP or DCX-R303X-GFP on the MT lattice. (**B**) Images of taxol MTs, 0.5 nM DCX-GFP, 2.5 nM DCX-mCherry, and the color-combined image of DCX-GFP and DCX-mCherry restricted to a mask generated from the taxol MT image. (**C**) Mean Pearson correlation coefficient for the mCherry and GFP intensities of the masked images. At least three fields of view were collected for each condition.

interactions between NDC and the MT lattice induce an opening of the CDC, leading to a binding event that favors the published crystallographic structure of the dimerized CDC (*Rufer et al., 2018*; *Figure 8*). The dimerization must be induced as it does not exist in solution without the MT interaction (*Moores et al., 2006*). The cryo-EM data seem to support this, as the conformational state of the CDC in the lattice-bound form is quite distinct from the X-ray structure of the free domain (*Manka and Moores, 2020*).

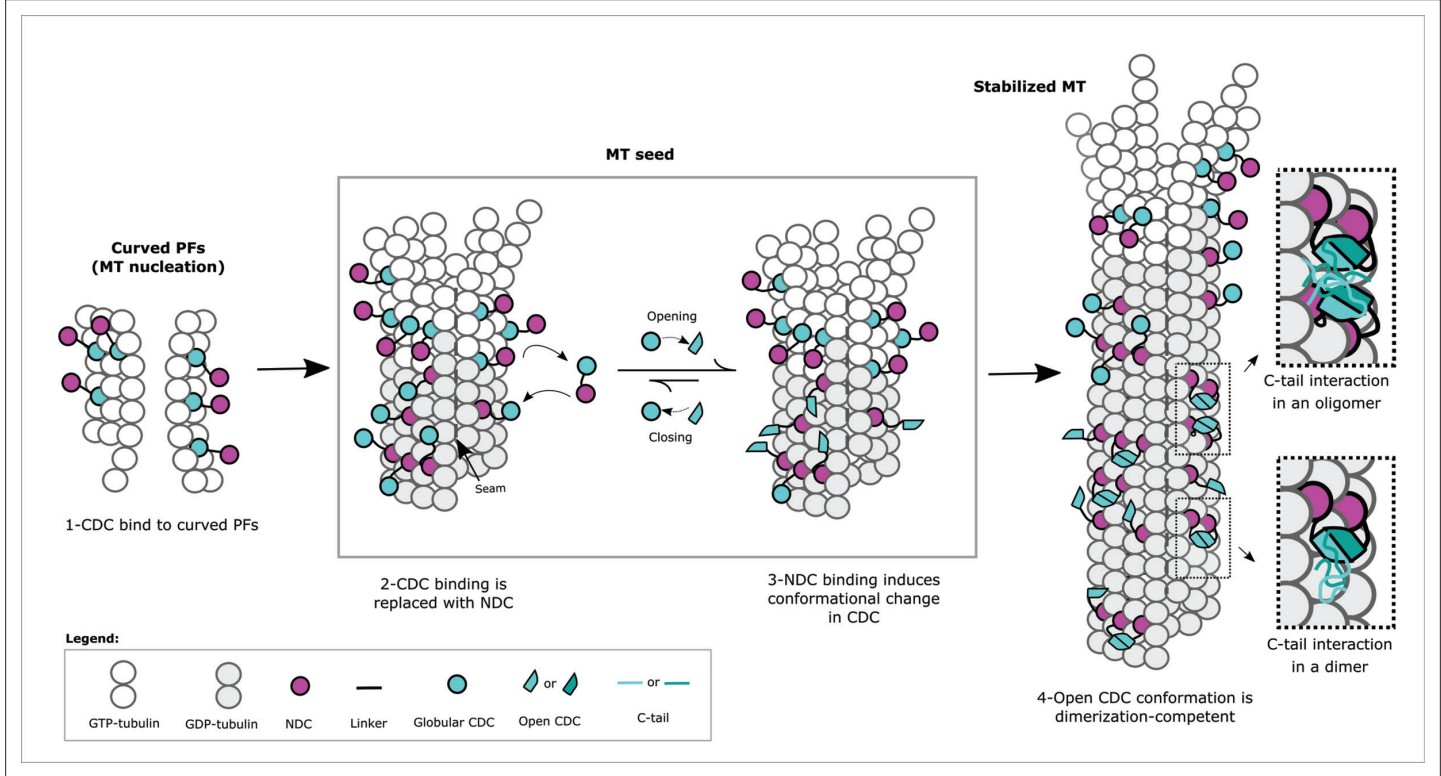

**Figure 8.** Mechanism of Doublecortin (DCX)-mediated microtubule (MT) nucleation and stabilization. DCX stabilizes early GTP-tubulin oligomers through the CDC domain, which is then replaced with the NDC domain at the canonical binding site as during full MT assembly. NDC binding triggers a conformational change in CDC domain, which facilitates DCX self-association and prevents CDC from rebinding. The inserts illustrate the role of C-tail domain in the formation of either intermonomer interactions, interdimer interactions, or both.

Induced dimerization provides an explanation for the transition from an early CDC–MT interaction to an NDC–MT interaction (*Manka and Moores, 2020*; *Figure 8*). It was proposed that CDC participates in nucleating MT growth when centrosomal γTURC complexes are not available, as is the case in the distal ends of migrating neurons (*Manka and Moores, 2020*). The deformable CDC domain creates junctional sites that may be somewhat different than the sites in the mature MT lattice because of the curved nature of the nucleated tubulin oligomers. As the lattice fully assembles and regular junctional sites are formed, the more rigid NDC domain can now bind. It is important to note that neither of the two structured domains bind to the lattice particularly strongly on their own, thus a transition to NDC binding could readily occur upon lattice maturation (*Manka and Moores, 2020*). The CDC interactions would diminish over time, provided that the structural features of the lattice prevent their rebinding, or a mechanism exists for their sequestering (or both). Our model suggests that sequestering is very likely to occur. Induced unfolding of CDC followed by its capture in a dimeric state would explain how DCX binds to the lattice so avidly and such an association would certainly prevent CDC from rebinding. At the same time, dimerization would stabilize the MT lattice and offer a compelling reason for favoring MTs with a well-defined number of protofilaments. That is, the more rigid NDC coupled with its larger footprint on the lattice (*Manka and Moores, 2020*) is better positioned than the CDC to set the angle of curvature, and dimerization increases the local concentrations of these domains on the lattice through avidity.

There are several functional observations that support the idea that dimerization helps define the overall MT architecture. A double NDC construct induces much more MT bundling than the WT (*Manka and Moores, 2020*), very likely through cross-lattice interactions, suggesting that dimerization minimizes these interactions. This is consistent with the conformation shown in *Figure 6B*. A monomeric form may allow for bridging events leading to bundling, but our dimerization model would sterically hinder them. Dimerization is also supported through an inspection of patient mutations. These mutations cluster in both the NDC and CDC domains and lead to loss of MT binding and stabilization. It was suggested to be evidence that both domains interact with MTs directly (*Taylor et al., 2000*; *Kim et al., 2003*). Mutations in the NDC could certainly lead to reduced binding (*Manka and Moores, 2020*), but the mutations in the CDC region would more likely interfere with dimerization, reduce the avidity of DCX on the lattice, and diminish MT stabilization overall (*Bechstedt and Brouhard, 2012*; *Bechstedt et al., 2014*).

Extended self-association involves the C-tail and additional observations support a role for this domain in regulating the MT architecture. Removing the C-tail reduces the preference for 13-protofilament MTs (*Moores et al., 2004*). Although its removal slightly increases DCX affinity to MTs, it also reduces the rate of DCX accumulation on the lattice (*Moslehi et al., 2017*). These findings suggest that the C-tail influences either the span of the dimer or the stability of the dimerization event. The depth of crosslinking we observed in this region suggests an induced structure, perhaps a coiled-coil as observed with other dimerized MAPs (*Slep et al., 2005*; *Kon et al., 2009*). However, DeepCoil (*Ludwiczak et al., 2019*) shows a very low probability of such a domain. A more extensive multi-DCX network backed by a dynamically interacting set of C-tails would contribute to the high cooperativity of binding and help stabilize DCX on the lattice (see insets in *Figure 8*). It is possible that the C-tails may influence the NDC–MT interaction in a more direct fashion but either way, the self-association of the C-tails influence the properties of DCX on the lattice. This effect can be regulated by C-tail phosphorylations (*Tanaka et al., 2004*; *Graham et al., 2004*; *Shmueli et al., 2006*). For example, phosphorylation of S297 by Cdk5 reduces DCX affinity to MTs in cultured neurons and any mutations (phosphomimics or dephosphomimics) similarly reduce affinity and alter neuronal migration patterns (*Tanaka et al., 2004*). Conversely, spinophilin-induced dephosphorylation of S297 by PP1 causes an increase MT bundling in the axonal shaft of neurons (*Bielas et al., 2007*). Collectively, these observations indicate that higher-order self-association and its regulation are important to DCX function on the MT lattice.

At this point, it is unclear if the precise orientation of the dimer on the MT lattice is important or not. DCX recognizes a specific tubulin spacing in the lattice, favoring longitudinal curvatures that result in shorter interdimer distances (*Ettinger et al., 2016*). It suggests that a longitudinal orientation is possible, but this particular property may be a function of direct CDC binding at curvatures (in monomeric form, *Figure 8*), which is consistent with a role in sensing MT plus-end geometries (*Bechstedt et al., 2014*; *Manka and Moores, 2020*). Missense mutations in the CDC region (and not NDC)

disrupt the longitudinal curvature detection properties of DCX, but this could influence dimerization as well. Our integrative modeling is not sufficiently precise to identify a dominant orientation (if one truly exists), but a lateral interaction seems the most likely given the orientation of the NDC–CDC linker in cryo-EM models (*Manka and Moores, 2020*). A dimer with an ability to engage the lattice in a lateral manner could reinforce a preference for a 13-protofilament lattice, if the 'bite length' of the dimer is complementary to protofilament spacing.

Finally, the success of this integrative modeling technique relies heavily upon an improved cross-linking methodology that samples the equilibrated state more faithfully (*Ziemianowicz et al., 2019*). The presence of nominally disordered regions in MAPs like DCX renders proteins prone to 'kinetic trapping' on the lattice, where conventional long-lived crosslinking reagents can trap interactions in non-native states. We conducted a modeling analysis of the DCX–MT interaction using typical DSS (disuccinimidyl suberate) and EDC(1-ethyl-3-(3-dimethylaminopropyl)carbodiimide) crosslinkers and while the data are abundant (*Figure 4—figure supplement 4*), the outcome of the integrative modeling using conventional crosslinking chemistries was poor, both in terms of the percent crosslink satisfaction (<50%) and the location of the primary binding site. The full modeling exercise with these data provided no interpretable results. While a small number of studies have applied crosslinking to MAP–MT interactions (*Legal et al., 2016*; *Zelter et al., 2015*; *Abad et al., 2014*; *Kadavath, 2015*), many more could be profiled with an improved crosslinking method such as we illustrate here.

# Materials and methods

## Key resources table

| Reagent type (species) or resource | Designation | Source or reference | Identifiers | Additional information |
|---|---|---|---|---|
| Gene (*Homo sapiens*) | DCX | UniProtKB | O43602 | |
| Gene (*Sus scrofa*) | α-Tubulin | UniProtKB | P02550 | |
| Gene (*Sus scrofa*) | β-Tubulin | UniProtKB | P02554 | |
| Strain, strain background (*Escherichia coli*) | Arctic Express (DE3) | Agilent | 230,192 | Electrocompetent cells |
| Strain, strain background (*Escherichia coli*) | BL21(DE3) | New England BioLabs Inc. | C2527 | Mix and Go competent cells |
| Antibody | Anti-His (Mouse monoclonal) | Applied Biological Materials | G020 | WB (1:1000) |
| Antibody | Anti-β-tubulin (Mouse monoclonal) | Sigma-Aldrich | T4026 | (1:20 dilution in BRB80) |
| Recombinant DNA reagent | DCX-WT-pHAT-HUS | Gift of Dr. Susanne Bechstedt | | Human doublecortin (1–365) plasmid |
| Recombinant DNA reagent | DCX-WT-pHAT-HUGS | Gift of Dr. Susanne Bechstedt | | GFP version of Human doublecortin (1–365) plasmid |
| Recombinant DNA reagent | DCX-R303X-pHAT-HUGS | Gift of Dr. Susanne Bechstedt | | GFP version of Human doublecortin (1–302) plasmid |
| Recombinant DNA reagent | DCX-WT-pHAT-HUCS | Gift of Dr. Susanne Bechstedt | | mCherry version of Human doublecortin (1–365) plasmid |
| Peptide, recombinant protein | Human doublecortin (1–365) | This paper | | Purified from *E. coli* Arctic Express cells |
| Peptide, recombinant protein | GFP-doublecortin (1–365) | This paper | | Purified from *E. coli* BL21 cells |
| Peptide, recombinant protein | GFP-doublecortin (1-302) | This paper | | Purified from *E. coli* BL21 cells |
| Peptide, recombinant protein | mCherry-doublecortin (1-365) | This paper | | Purified from *E. coli* BL21 cells |
| Peptide, recombinant protein | α/β-Tubulin | Cytoskeleton | TL590M-A | |
| Peptide, recombinant protein | Rhodamine-labeled α/β-tubulin | Cytoskeleton | T240 | |

*Continued on next page*

*Continued*

| Reagent type (species) or resource | Designation | Source or reference | Identifiers | Additional information |
|---|---|---|---|---|
| Peptide, recombinant protein | Streptavidin–HRP | Thermo Fisher Scientific | N100 | |
| Peptide, recombinant protein | Glucose oxidase | Sigma-Aldrich | G2133-10KU | |
| Peptide, recombinant protein | Catalase | Sigma-Aldrich | E3289 | |
| Chemical compound, drug | Paclitaxel | European Pharmacopoeia Reference Standard | Y0000698 | |
| Chemical compound, drug | Docetaxel | Sigma-Aldrich | 01885 | |
| Chemical compound, drug | Atto 633 NHS-ester | ATTO-TEC GmbH | AD 633-35 | |
| Chemical compound, drug | LC-SDA | Thermo Fisher Scientific | 26,168 | |
| Chemical compound, drug | DSS | Thermo Fisher Scientific | 21,655 | |
| Chemical compound, drug | EDC | Thermo Fisher Scientific | 22,980 | |
| Software, algorithm | xVis | https://xvis.genzentrum.lmu.de/login.php | | PMID:25956653 |
| Software, algorithm | xiNET | http://crosslinkviewer.org/ | | PMID:25648531 |
| Software, algorithm | IMP | https://integrativemodeling.org/ | v.2.12 | PMID:22272186 |
| Software, algorithm | Mass Spec Studio | https://www.msstudio.ca | V2.0 | PMID:25242457 |
| Software, algorithm | TrackMate | https://imagej.net/plugins/trackmate/ | | PMID:27713081 |

## Expression and purification of DCX

The production method for DCX was described previously (*Bechstedt and Brouhard, 2012*). Briefly, the plasmid for human DCX (sp|O43602|DCX_HUMAN/1–365) includes an N-terminal polyhistidine (6His) tag and a StrepTag II (DCX-pHAT-HUS). Additional plasmids contained these tags as well as a C-terminal GFP label (DCX-pHAT-HUGS) or a C-terminal mCherry label (DCX-pHAT-HUCS). A fourth plasmid for the 303X mutant includes an N-terminal polyhistidine (6His) tag and a StrepTag II, and a C-terminal GFP label (DCX-R303X-pHAT-HUGS). Plasmids were transfected and grown in *E. coli* BL21. Briefly, 100 ml 2YT was supplemented with 100 µg/ml ampicillin and 20 µg/ml gentamycin at 32°C overnight and cells grown to OD600 ~1–3. Cells were resuspended in fresh media and grown at 37°C to mid-log phase (OD600 ~0.6–1.0). DCX expression was induced by addition of IPTG(Isopropyl ß-D-1-thiogalactopyranoside) and incubated for 16 hr at 18°C. Cells were harvested by centrifugation and stored at −80°C until use.

For the expression of isotopically labeled DCX, DSB126 (DCX-pHAT-HUS) was grown as above to $OD_{600}$ ~1–3, using Arctic Express *E. coli*. The rich 2YT media was removed and cells resuspended in 1 l of freshly prepared $^{15}$N-M9 minimal media (*Lima et al., 2018*), with 100 µg/ml ampicillin and 20 µg/ml gentamycin. The cells were then grown at 37°C to mid-log phase ($OD_{600}$ ~0.8), the temperature reduced to 16°C for 1 hr, then induced with 1 mM IPTG overnight. Cells were harvested by centrifugation and stored at −80°C until use.

For protein purification, for either state, DCX-expressing cells were thawed and resuspended in cold lysis buffer (50 mM Tris–HCl pH 8, 300 mM NaCl, 10 mM imidazole, 10% glycerol, complete protease inhibitor, 1 mM PMSF(phenylmethylsulfonyl fluoride)) and lysed by sonication (Qsonica, Newtown, USA) on ice. After centrifugation, the supernatant was loaded onto a HisTrap column (5 ml, Cat. No. 17-5248-01, GE Healthcare) pre-equilibrated with lysis buffer. The column was washed with His-Buffer A (50 mM Tris–HCl pH 8, 300 mM NaCl, 10 mM imidazole, 1 mM PMSF) a protein eluted with a gradient of 0–100% His-Buffer B (50 mM Tris–HCl pH 8, 300 mM NaCl, 250 mM imidazole, 1 mM PMSF). Fractions were collected and analyzed by western blot, probed with anti-His (Applied Biological Materials Inc, Richmond, Canada) and streptavidin–HRP(horseradish peroxidase) (Thermo Fisher Scientific). DCX-containing fractions were dialyzed (MWCO 6–8 kDa) in Strep-Buffer A (100 mM Tris–HCl pH 8, 150 mM NaCl, 1 mM EDTA) at 4°C overnight. The dialyzed DCX sample was centrifuged to remove any precipitates before loading onto a Strep-Trap (1 ml, Cat. No. 29-0486-53, GE Healthcare) column. After washing with Strep-Buffer A, DCX was eluted with a gradient of 0–100% Strep-Buffer

B (100 mM Tris–HCl pH 8, 150 mM NaCl, 1 mM EDTA, 2.5 mM D-desthiobiotin, 10% glycerol [VWR Life Science]). Fractions were analyzed by western blot as above and DCX-containing fractions were dialyzed in BRB80 (80 mM PIPES(piperazine-N,N′-bis(2-ethanesulfonic acid)), 1 mM EGTA(ethylene glycol-bis(β-aminoethyl ether)-N,N,N′,N′-tetraacetic acid), 1 mM MgCl$_2$, pH 6.8) and supplemented with 1 mM GTP (Enzo Life Sciences, Farmingdale, USA). After centrifugation, the supernatant was concentrated on an Amicon MWCO 10 kDa unit (Millipore). The concentration of DCX was determined by Nanodrop (Thermo Fisher).The light-DCX protein used in this study was prepared in two separate batches (one for method optimization) and the heavy-DCX from one batch.

## Fluorescence microscopy

Porcine Tubulin at 1:5 ratio (rhodamine-labeled: unlabeled) (Cat. No. T240 and TL590M-A, respectively, Cytoskeleton, Inc, Denver, USA) was reconstituted to 4.0 mg/ml in cold polymerization buffer, BRB80. Centrifugation was performed at 16,800 × $g$ for 10 min at 4°C to pellet any aggregates. The supernatant was diluted in warm BRB80 to 10 µM (1.0 mg/ml) and supplemented with 10 µM DCX and incubated at 37°C for 90 min. The resulting DCX–MT constructs were diluted 100 times in warm resuspension buffer (BRB80, 10 µM Docetaxel) immediately before fluorescence microscopy. Diluted sample was placed on a glass slide and covered with cover glass (glass slides were acid-etched overnight before use and rinsed thoroughly). MTs were imaged using an AxioObserver epifluorescence microscope with an oil immersion objective (EC Plan Apo ×100/1.3, Carl Zeiss MicroImaging GmbH, Jena, Germany) equipped with CCD camera Zeiss AxioCam MRm Rev. 3 FireWire. Images were taken using 100% light source intensity and 4-s exposure time. Microscopy images were processed to adjust for contrast using Zen Blue software (version 2.3). MT length measurement was performed using standard measurement tools in ImageJ (*Schindelin et al., 2012*).

## Turbidity assay

Tubulin (10 µM) and different concentrations of DCX (0–20 µM) were premixed in BRB80 + GTP (1 mM) on ice and transferred to a 396-well plate. The plate was transferred to the temperature controlled (37°C) spectrophotometer (Molecular Devices FilterMax F5 spectrophotometer equipped with a 340-nm filter). Turbidity measurements were taken every 13 s for over 70 min. Data were normalized to a preliminary time point and the pathlength was corrected to 1 cm.

## Cooperativity assay

Taxol MTs were prepared from bovine tubulin as previously described (*Bechstedt et al., 2014*). Briefly, on the day of each experiment, taxol MTs were prepared by polymerizing a 1:30 molar ratio of Atto 633 NHS-ester (ATTO-Tec, AD 633-35) labeled:unlabeled tubulin. The cover glass was cleaned in acetone, 50% methanol, and 0.5 M KOH, then exposed to air plasma for 3 min at 30–40 cc/min, finally silanized by soaking in 0.1% Dichlorodimethylsilane (Sigma-Aldrich, 440272) in *n*-heptane for at least 3 hr. Sample chambers were assembled to create flow channels for solution exchange as described previously (*Gell et al., 2010*). Channels were prepared by flowing in anti-β-tubulin (Sigma-Aldrich, T4026) followed by blocking with 1% Pluronic F-127. Channels were rinsed with 10 channel volumes of BRB80 before flowing in taxol MTs and placing the chamber on the microscope stage, where the objective was heated to 32°C with a CU-501 Chamlide lens warmer (Live Cell Instruments). DCX-mCherry and DCX-GFP were mixed on ice in imaging buffer (BRB80, 0.1 mg/ml bovine serum albumin, 10 mM dithiothreitol, 250 nM glucose oxidase, 64 nM catalase, 40 mM D-glucose, 0.01% (wt/vol) methylcellulose, and 10 µM paclitaxel) before flowing into the channel and sealing with nail polish. For each field of view, all three signals (GFP, mCherry, and Atto 633) were imaged at an exposure time of 1 s and with wavelength-specific filter cubes (Chroma). At least three fields of view were imaged for each concentration of DCX-mCherry. Images were recorded on a Prime 95B CMOS camera (Photometrics) with a pixel size of 107 nm.

## Analysis of DCX cooperativity

To correct for chromatic aberrations, images were processed in FIJI by applying a 0.5 px median and subtracting their 10 px Gaussian-filtered form. The three images of a single field of view (DCX-GFP, DCX-mCherry, and Atto-633 taxol MTs) were aligned using the positions of fiducial particles detected in TrackMate. For each field of view, the Atto-633 taxol MT signal was used to generate mask by Otsu

thresholding in FIJI, ignoring pixels outside of the MT region defined by the mask. We quantified the colocalization of DCX-mCherry and DCX-GFP by measuring the Pearson correlation coefficient of the mCherry and GFP signal along the MTs, using the scipy.stats.pearsonr function.

## Crosslinking

Porcine Tubulin was reconstituted to 4.0 mg/ml in cold BRB80 polymerization buffer supplemented with 1 mM GTP. Centrifugation was performed at 16,800 × $g$ for 10 min at 4°C to pellet any aggregates. The supernatant was diluted to 1.0 mg/ml (10 μM) in the presence of 10 μM DCX and incubated at 37°C for 90 min to induce polymerization. Succinimidyl 6-(4,4'-azipentanamido)hexanoate (LC-SDA; Thermo Scientific) crosslinking was performed by adding the reagent to final 1 mM concentration. The sample was incubated for 10 min at 37°C, followed by 5 s of photolysis at 355 nm (50 × 100 mJ laser pulse of a 10-ns pulse width using an Nd:YAG laser, YG 980; Quantel, Les Ulis, France) (*Ziemianowicz et al., 2019*). DSS crosslinking was performed using 1 mM crosslinker concentration followed by 30-min incubation at 37°C. EDC crosslinking was performed with the same crosslinker concentration and incubation time, in the presence of 2 mM Sulfo-NHS. All chemicals were purchased from Sigma-Aldrich, unless mentioned otherwise. The crosslinking experiment was repeated several times for the purpose of optimization. Final results are the accumulated identifications across two replicate analyses.

## Mass spectrometry

Crosslinked DCX–MT was separated from free protein by centrifugation. The supernatant was removed, and the pellet washed once with warm BRB80 (37°C) and then dissolved in 50 mM ammonium bicarbonate solution to a final protein concentration of 1 mg/ml. Cysteines were reduced by adding DTT to a final concentration of 10 mM with heating to 90°C for 10 min, and then alkylation by addition of chloroacetamide to a final concentration of 50 mM at 37°C for 30 min. Trypsin (Thermo Scientific) was added to an enzyme-to-substrate ratio of 1:50 and incubated overnight at 37°C with nutation (150 rpm). Digestion was quenched by adding formic acid to a final concentration of 2%. Samples were aliquoted and either lyophilized for size-exclusion chromatography (SEC) or desalted using ZipTip C18 pipette tips (Merck Millipore Ltd, Ireland) for LC–MS/MS analysis. For SEC, samples were reconstituted in SEC buffer consisting of 30% acetonitrile (LC–MS grade, Thermo Scientific) and 0.1% FA (Fluka) and crosslinked peptides were enriched by separation on a Superdex Peptide PC 3.2/30 column (GE Healthcare), using an Agilent 1100 chromatography system at 95 μl/min with fraction collection. Fractions were lyophilized. SEC fractions and unenriched samples were reconstituted in 0.1% formic acid and based on the UV absorbance trace, approximately 1 pmol injected onto an Acclaim PepMap 100 guard column (75 μm × 2 cm C18, 3 μm particles, 100 Å; Thermo Scientific) and separated on a 50 cm PepMap RSLC C18 (75 μm × 50 cm, 2 μm particles, 100 Å; Thermo Scientific) coupled to an Orbitrap Fusion Lumos (Thermo Scientific). Peptides were separated with a gradient consisting of 56 min at 5–28% B, 17 min at 28–40% B, followed by 10 min 40–95% B and regenerated at 95% B for 10 min. Mobile phase A was 0.1% FA in water, and mobile phase B was 0.1% FA in 80% I. The flow rate was 300 nl/min. The mass spectrometer was operated in positive ion mode, in OT/OT mode with MS resolution set at 120,000 (350–1300 *m/z*) and MS2 resolution at 15,000. The max injection time was set at 50 and 100 ms for MS1 and MS2 scans, respectively. The AGC target was set at 400,000 and 100,000 for MS1 and MS2, respectively. Higher-energy collisional dissociation was used to generate MS2 spectra, for charge states 4+ and higher. A normalized collision energy of 32% was used with an isolation width of 1.2 *m/z*.

## DCX dimerization analysis

[14]N-DCX (light) and [15]N-DCX (heavy) were mixed in 1:1 ratio and then applied to tubulin as above. To explore dimerization and/or aggregation, the light–heavy mixture was processed using a few different approaches: (1) mixture incubated at 10 μM for 2 hr at 37°C with tubulin, (2) mixture incubated at 10 μM for 2 hr at 37°C without tubulin, (3) mixture diluted to 2 μM followed by concentration to ≥10 μM and incubation at 37°C for 2 hr without tubulin, (4) mixture denatured in 3 M guanidyl-HCl in phosphate-buffered saline overnight followed by buffer exchange into BRB80 and concentration to ≥10 μM and incubation at 37°C for 2 hr without tubulin. A negative control sample was also prepared by crosslinking heavy and light DCX separately and mixing them immediately prior to LC–MS/MS

analysis. All samples were crosslinked by LC-SDA and processed as above, using only the nonenriched samples.

## XL-MS data analysis

Raw LC–MS/MS data were imported into CRIMP v2 (the crosslinking plugin in the Mass Spec Studio, https://www.msstudio.ca) (*Sarpe et al., 2016*) along with the Fasta files of the Tubulin isoforms identified previously (TB-α–1A, TB-α–1B, TB-α–1C, TB-α–1D, TB-α–4A, and TB-β, TB-β–2B, TB-β–4A, TB-β–4B, TB-β–3, TB-β–5) (*Rafiei and Schriemer, 2019*). The minor impurities in the recombinantly purified DCX did not have an impact on crosslinking results. For crosslinked peptide searching, methionine oxidation and carbamidomethylation of cysteines were selected as variable and fixed modifications, respectively. Crosslinked peptides were searched using the following parameters: MS accuracy, 5 ppm; MS/MS accuracy, 10 ppm; E-threshold, 70; enzyme, trypsin (K/R only); $m/z$ range: 350 and 1300; peptide length range: 3–50 residues. All crosslinking results filtered at an estimated 1% FDR(false discovery rate) level were manually validated. Monoisotopic ion identification in MS1 was confirmed in all hits and only crosslinked peptides with a good fit of the measured isotopic envelope to the theoretical isotopic pattern were accepted. Crosslinking data were exported as CSV files and data from all SEC fractions combined. For isotopically labeled proteins, CRIMP was customized to search all heavy–light peptide combinations.

## DCX–MT integrative structure modeling

We used integrative structure modeling (ISM) to assess how our generated crosslinking data, along with the previously computed EM map, prior structural information and physical principles, could model the interaction between the domains of DCX and the MT lattice. Briefly, data and information were transformed into spatial restraints used to construct a scoring function and then we searched for model configurations that satisfy all restraints. To assess whether NDC or CDC binds to MT, we performed ISM using the MT lattice and either CDC or NDC along with intermolecular crosslinks between each domain and MT. The convergence of each of the DCX domains to the observed binding site in the EM map using only the crosslinking data was analyzed. In addition, the satisfaction of the crosslinking restraint was computed while enforcing a restraint using the EM density of the domain of DCX bound to MT to guide the domain to this observed site, with the better models identified by those that better satisfy the crosslinks.

We assessed the nature of the DCX dimerization event by performing ISM using two DCX molecules and a 3–4 repeat (lateral) and 3–4 repeat (longitudinal) MT domain constructed using tubulin and symmetry operations. The position of NDC (NDC structure of PDB: 4ATU *Liu et al., 2012*) was fixed on the MT lattice in the site observed by EM. CDC was modeled in both the open (domain-swappable, PDB 6FNZ *Rufer et al., 2018*) and closed (globular structure, PDB: 5IP4 *Burger et al., 2016*) forms. Models were scored via the set of all DCX–DCX crosslinks and models were assessed using crosslink satisfaction. Finally, we evaluated the ability of DCX dimers to be formed using four spatial arrangements of DCX dimers on the MT lattice. Two copies of NDC were fixed in adjacent binding sites lateral, longitudinal, diagonal-1, and diagonal-2. Models were generated using restraints based on all crosslinking data (DCX–DCX and DCX–MT).

ISM protocols were built using the Python Modeling Interface (PMI) package of the open-source Integrative Modeling Platform (IMP) package (*Webb et al., 2018*) version 2.10 (https://integrative-modeling.org). Specifically, ISM proceeds through four stages: (1) gathering data, (2) representing subunits and translating data into spatial restraints, (3) sampling configurations of the system representations to produce an ensemble of models that satisfies the restraints, and (4) analyzing and validations the ensemble structures (*Figure 3*).

## Stage 1: gathering data

The 124 unique crosslinks identified between DCX and α–β-tubulin, along with 74 unique inter- and intra- DCX crosslinks identified in this study informed the spatial proximities of the CDC and NDC with respect to MT lattice and each other. An EM map containing MT and DCX (EMDB 2095; *Liu et al., 2012*) at 8.0 Å resolution was used to inform the binding mode of DCX with respect to the MT lattice. The map was prepared as follows. First, the cryo-EM structure for MT polymerized in the presence of DCX (6EVZ *Manka and Moores, 2018*) was fitted into the EM map using UCSF Chimera

tools. Second, the EM volumes corresponding to the MT structure were erased using Erase tool in Chimera, leaving only the DCX density. The representation of the MT lattice was derived from the high-resolution structure built based on the cryo-EM structure for MT polymerized in the presence of DCX (PDB: 6EVZ *Manka and Moores, 2018*). The representation of the NDC globular domain was informed by the PDB structure 4ATU (*Liu et al., 2012*) and the representation of the globular domain of CDC informed by either the crystal structures of the globular (PDB 5IP4; *Burger et al., 2016*) and domain-swapped (PDB 6FNZ; *Rufer et al., 2018*) configurations.

## Stage 2: system representation and translation of data into spatial restraints

The information above was used to define the representation of the system, the scoring function that guides the search for configurations that satisfy the information, filtering of models and validation of the final ensembles. One representation of the MT lattice was constructed based on the coordinates of the PDB structure (6EVZ; *Manka and Moores, 2018*), with components simultaneously represented as beads of a single residue each and up-to-10 residues-per-bead. An additional 4-dimer width MT lattice representation was made using the same PBD structure (6EVZ) of MT lattice and symmetry replication tools in UCSF Chimera (*Kim et al., 2003*).

Residues of NDC (51–140) and CDC (174–251 for globular and 174–253 for open form) contained in the crystal structure (structured domains) were represented simultaneously as beads of up-to-10 residues-per-bead. Residues not contained in the crystal structures (terminal tails and the linker region between NDC and CDC) were modeled solely as beads of up-to-10 residues each. Both the NDC and CDC structured domains were constrained as rigid bodies: the intermolecular distances between these beads were kept fixed. All other beads were unconstrained. The N-terminal (1–50) and C-terminal (331–360) tails of DCX were not modeled, as these regions were less represented in inter-and/or intra-DCX XL sites. For different modeling protocols, either one of the NDC/CDC was modeled, or two copies of DCX modeled.

The information was then encoded into spatial restraints computable on the system representation. Our scoring function consisted of the sum of four component restraints. (1) The *chemical crosslink restraint* utilized the identified chemical crosslinks to construct a Bayesian scoring function (*Gutierrez et al., 2020*) that restrained distances between crosslinked residues. Restraint distances of 30 Å for LC-SDA and DSS, and 25 Å for EDC were used. The ambiguity of crosslink sites due to the presence of multiple copies of the same protein was considered. In these cases, the joint probability of satisfying the restraint over all possible combinations is computed (*Molnar et al., 2014*). (2) The *EM restraint* utilized a Bayesian scoring function based on the cross-correlation of the overlap of the model and experimental density. Densities of the model components and experimental data were approximated using a Gaussian Mixture Model (*Bonomi et al., 2019*). (3) The *excluded volume restraint* prevents parts of the system from occupying the same space. This restraint is applied to the low-resolution representation of the system (10 residues per bead). (4) The *sequence connectivity restraint* was used to restraint chemical components known to be covalently linked. The restraint was applied as a harmonic upperdistance bound on the distance between adjoining beads in sequence. The center of the harmonic was defined as twice the sum of the radii of the two beads, the radii computed from the excluded volume of the bead.

## Stage 3: configurational sampling

The search for model configurations that satisfy the restraints used Gibbs sampling, based on the Metropolis Monte-Carlo algorithm and accelerated via replica exchange. Initial positions of system components were randomized with exception of MT being fixed for all cases, and NDCs being fixed for dimeric-DCX modeling scenarios. The set of Monte-Carlo moves consisted of random rotation and translation of rigid bodies (up to 4.0 Å and 0.5 radians, respectively) and random translations of the individual beads not in rigid bodies of up to 4 Å. Model configurations were saved every 10 steps. For each modeling protocol, 8–10 independent sampling runs were initiated using 20–40 replicas and run for 100,000–500,000 steps each, resulting in 100,000–500,000 models for each protocol.

## Stage 4: analyzing and validating model ensembles

The resulting ensemble of model configurations were analyzed to estimate the structural precision, ensure appropriate consistency with the input data and suggest more informative future experiments. The models were first filtered for those that satisfy the input data. After filtering, we used analysis and validation protocols published previously (*Ishida and Kinoshita, 2007*). Briefly, analysis began by assessing the thoroughness of structural sampling and computation of the sampling precision, the RMSD threshold at which clustering produces indistinguishable results for two independent sets of computed models (e.g., the output of runs 1–5 and 6–10), using protocols previously described (*Viswanath et al., 2017*). Upon computation of the sampling precision, the entire set of models was clustered at this precision or higher to ensure that resulting model clusters reflect the uncertainty in sampling in input information. The centroid models of clusters were then filtered to remove the models where NDC/CDC is located on the MT edges (where another tubulin monomer will fit in a full MT structure). These model clusters were further assessed by computing their precision (structural variability – average RMSD to the centroid model) and quantifying their fit to the input information.

These final models were validated by ensuring that the models satisfied the crosslinking data used to compute it. Crosslink satisfaction was also used to determine which modeling configurations were more plausible than others. The fraction crosslink satisfaction was computed both for each individual model configuration in a cluster and the entire cluster. A crosslink was deemed to be satisfied if the crosslink distance in an individual model configuration was less than 35 Å for LC-SDA and DSS and 30 Å for EDC. A crosslink was deemed to be satisfied in a model cluster (called cluster crosslink satisfaction in Table S2 in *Supplementary file 2*) if at least one configuration in the cluster has a crosslink distance less than the distance thresholds described above. Also, RMSD calculation was performed between each individual model present in the clusters and the NDC component of PDB structure 4ATU. RMSD calculation for CDC–MT models was performed using an aligned globular CDC structure (PDB 5IP4) into the NDC component of cryo-EM based structure (PDB 4ATU). The full details of sampling precision, clustering threshold, cluster's population, etc. for all modeling scenarios are presented in Table S2 in *Supplementary file 2*.

## Data availability

The DCX–MT integrative models, including final structures, modeling details, and input experimental data, were deposited into the PDB-dev repository for integrative models (https://www.pdb-dev.com) accession number PDBDEV_00000071, PDBDEV_00000072, PDBDEV_00000073, and PDBDEV_00000074. All LC–MS/MS data generated to support the findings of this study have been deposited with the ProteomeXchange Consortium via the PRIDE (*Perez-Riverol et al., 2019*) partner repository with the dataset identifier PXD033167.

## Acknowledgements

We thank Dr. Laurent Brechenmacher for consultation in LC–MS/MS method development. We thank Dr. S Bechstedt for the gift of the DCX expression constructs. This work was supported by CANARIE (project RS-326) and NSERC (RGPIN-2017-04879). G. Brouhard was supported by the Canadian Institutes of Health Research (PJT-148702) and NSERC (RGPIN-2020-04876). C. Edrington was supported by an NSERC PGS-D postgraduate scholarship. S. Cruz Tetlalmatzi was supported by a McGill Engineering Doctoral Award.

## Additional information

### Funding

| Funder | Grant reference number | Author |
|---|---|---|
| Canarie | RS-326 | David C Schriemer |
| Natural Sciences and Engineering Research Council of Canada | RGPIN-2017-04879 | David C Schriemer |

| Funder | Grant reference number | Author |
|---|---|---|
| Natural Sciences and Engineering Research Council of Canada | RGPIN-2020-04876 | Gary Brouhard |
| Canadian Institutes of Health Research | PJT-148702 | Gary Brouhard |

The funders had no role in study design, data collection, and interpretation, or the decision to submit the work for publication.

### Author contributions

Atefeh Rafiei, Formal analysis, Investigation, Methodology, Writing – original draft, Writing – review and editing; Sofía Cruz Tetlalmatzi, Formal analysis, Methodology, Writing – review and editing; Claire H Edrington, Formal analysis, Writing – review and editing; Linda Lee, Formal analysis, Investigation; D Alex Crowder, Software, Writing – review and editing; Daniel J Saltzberg, Andrej Sali, Methodology, Software, Visualization, Writing – review and editing; Gary Brouhard, Methodology, Software, Writing – review and editing; David C Schriemer, Conceptualization, Funding acquisition, Methodology, Supervision, Writing – review and editing

### Author ORCIDs

Andrej Sali http://orcid.org/0000-0003-0435-6197
Gary Brouhard http://orcid.org/0000-0001-9101-1247
David C Schriemer http://orcid.org/0000-0002-5202-1618

### Decision letter and Author response

Decision letter https://doi.org/10.7554/eLife.66975.sa1
Author response https://doi.org/10.7554/eLife.66975.sa2

## Additional files

### Supplementary files

• Supplementary file 1. Complete list of crosslinking sites (and their associated crosslinked peptides) identified for MT-DCX construct using LC-SDA photo-chemical crosslinking.

• Supplementary file 2. The analysis output of all integrative structural modeling runs. Column titles with star are defined at the bottom of the table.

• Supplementary file 3. Complete list of unique DCX-DCX crosslinking sites (and their associated crosslinked peptides) identified using isotope-assisted chemical crosslinking.

• Transparent reporting form

### Data availability

The DCX-MT integrative models, including final structures, modeling details, and input experimental data, were deposited into the PDB-dev repository for integrative models (https://www.pdb-dev.com) as follows: Dimeric DCX-MT (diagonal1): PDBDEV_00000071 Dimeric DCX-MT (lateral): PDBDEV_00000072 Dimeric DCX-MT (longitudinal): PDBDEV_00000073 Dimeric DCX-MT (diagonal2): PDBDEV_00000074 All LC-MS/MS data generated to support the findings of this study have been deposited with the ProteomeXchange Consortium with the dataset identifier PXD033167.

The following datasets were generated:

| Author(s) | Year | Dataset title | Dataset URL | Database and Identifier |
|---|---|---|---|---|
| Atefeh Rafiei and David C Schriemer | 2021 | DCX-MT crosslinking | https://www.ebi.ac.uk/pride/archive/projects/PXD033167 | PRIDE, PXD033167 |

*Continued*

| Author(s) | Year | Dataset title | Dataset URL | Database and Identifier |
|---|---|---|---|---|
| Rafiei A, Schriemer DC | 2021 | Dimeric DCX-MT (diagonal1) | https://pdb-dev-beta.wwpdb.org/entry.html?PDBDEV_00000071 | RCSB Protein Data Bank, PDBDEV_00000071 |
| Rafiei A, Schriemer DC | 2021 | Dimeric DCX-MT (lateral) | https://pdb-dev-beta.wwpdb.org/entry.html?PDBDEV_00000072 | RCSB Protein Data Bank, PDBDEV_00000072 |
| Rafiei A, Schriemer DC | 2021 | Dimeric DCX-MT (longitudinal) | https://pdb-dev-beta.wwpdb.org/entry.html?PDBDEV_00000073 | RCSB Protein Data Bank, PDBDEV_00000073 |
| Rafiei A, Schriemer DC | 2021 | Dimeric DCX-MT (diagonal2) | https://pdb-dev-beta.wwpdb.org/entry.html?PDBDEV_00000074 | RCSB Protein Data Bank, PDBDEV_00000074 |

The following previously published datasets were used:

| Author(s) | Year | Dataset title | Dataset URL | Database and Identifier |
|---|---|---|---|---|
| Liu JS, Schubert CR, Fu X, Fourniol FJ, Jaiswal JK, Houdusse A, Stultz CM, Moores CA, Walsh CA | 2012 | Structure of NDC | https://www.rcsb.org/structure/4ATU | RCSB Protein Data Bank, 4ATU |
| Rufer AC, Kusznir E, Burger D, Stihle M, Ruf A, Rudolph MG | 2018 | Domain-swapped structure of CDC | https://www.rcsb.org/structure/6FNZ | RCSB Protein Data Bank, 6FNZ |
| Burger D, Stihle M, Sharma A, Di Lello P, Benz J, D'arcy B, Debulpaep M, Fry D, Huber W, Kremer T | 2016 | Closed structure of CDC | https://www.rcsb.org/structure/5IP4 | RCSB Protein Data Bank, 5IP4 |
| Liu JS | 2012 | DCX-MT EM map | https://pdbj.org/emnavi/quick.php?id=2095 | EMDB, 2095 |
| Manka SW, Moores CA | 2018 | DCX-MT structure | https://www.rcsb.org/structure/6evz | RCSB Protein Data Bank, 6EVZ |

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
