## [Editor Report]

In their manuscript, Rafiei et al., investigate the molecular architecture of the microtubule-associated protein, doublecortin-X, by integrating data from chemical cross-linking experiments and in vitro molecular analysis with available crystallographic and cryo-EM structures. They determine the contribution of individual domains of this protein to microtubule-binding and self-association, providing a molecular framework for how this protein binds cooperatively along the microtubule lattice.

---

## [Decision Letter]

**Decision letter after peer review:**

Thank you for submitting your article “Doublecortin engages the microtubule lattice through a cooperative binding mode involving its C- terminal domain” for consideration by *eLife*. Your article has been reviewed by 3 peer reviewers, one of whom is a member of our Board of Reviewing Editors, and the evaluation has been overseen by Anna Akhmanova as the Senior Editor. The following individuals involved in review of your submission have agreed to reveal their identity: Carolyn A Moores (Reviewer #2); Alexander Leitner (Reviewer #3).

Essential revisions:

1) How accessible are the well-buried DCX-tubulin interfaces at the primary binding site to the chemical cross-linkers on which the analysis depends? As one reviewer points out, accessibility issues could explain the results depicted in Figure 3A, B, in which modelling that relies strictly on cross-links places NDC towards the outer edge of the protofilament, whereas inclusion of cryo-EM data in the integrated model places NDC in the inter-protofilament valley. The authors should discuss this further in the text and/or provide additional details about the measures they took to address accessibility issues.

2) It is important to repeat this analysis using the microtubule-bound structure of CDC (6RF2), given that this structure is conformationally distinct from PDB:5IP4. Based on analysis using the nanobody-bound CDC structure (5IP4), CDC appears to behave distinctly compared to NDC, such that CDC-derived cross-linking data are not consistent with the canonical inter-protofilament binding site. The authors should address if this depends on the particular PDB used.

3) Two reviewers noted that additional experiments would help support the observation and model that DCX oligomerises between the CDC domain and the C-terminal tail of the protein. The authors could either perform the crosslinking experiments with a previously published NDC-NDC chimera (Manka and Moores, 2020), which binds microtubules at the same inter-protofilament site but which lacks the CDC domain, or a DCX construct that lacks the C-terminal tail domain to examine if or how the interactions change.

4) Certain technical details and quantification must be clarified or reported within the manuscript:

The authors should indicate how many times were the crosslinking experiments repeated, and if they used more than one protein purification, which is ideal. It is clear in the methods that the authors do not perform column chromatography to ensure homogeneity of the protein. It is important to provide evidence that the protein is well-behaved using size-exclusion chromatography, for example. SEC-MALS would be ideal to ensure there are not multiple oligomeric populations of DCX in their preps, but if the authors do not have access to this equipment, normal SEC is adequate.

The authors should provide quantification of S1C, especially if they are assuming a 1:1 stoichiometry. Please report how many times was this binding assay repeated. Please also report the average lengths +/- s.d. for S1D. In addition, please provide zoom-in pictures of S1E for clear visualization of the microtubules. Finally, do the authors think that varying salt concentration in their buffer during their crosslinking studies would affect oligomerisation of DCX? The authors should provide an explanation for why they chose to perform these studies in BRB80 and indicate whether they think varying the ionic strength would affect the inter-molecular interactions.

5) The authors should discuss the results from Bechstedt and Brouhard 2012, which show that mutations in the C-DC disrupt cooperative interactions of DCX on the microtubule lattice. This study supports the authors’ current model greatly, yet it’s not discussed at all.

*Reviewer #1 (Recommendations for the authors):*

1) The authors should provide quantification of S1C, especially if they are assuming a 1:1 stoichiometry. How many times was this binding assay repeated? The average lengths are not reported in the manuscript for S1D. Provide zoom-in pictures of S1E, since it is impossible to see the microtubules in this image.

2) How many times were the crosslinking experiments repeated? Did the authors use more than one protein purification?

3) In Figure S6, the authors show that only under extreme conditions, such as denaturation, can DCX self-associate in the absence of tubulin. It appears as if the authors performed their crosslinking studies in BRB80 without additional salt. Are the authors able to perform the crosslinking studies at a more physiological salt concentration? Or could the authors provide an explanation as to why they do not think this would affect the inter-molecular interactions?

4) It would be useful to test the model generated in this study by performing certain experiments with a DCX construct that lacks the C-terminal tail. It is obviously difficult to evaluate microtubule binding without either DC domain, but if a tail-less DCX is still able to bind to microtubules, then it could be very useful in providing support for this model. Would a tail-less DCX exhibit less cooperative binding? Would removing the tail abolish the inter-molecular interactions? Or would the C-DC domains still be able to self-associate?

5) It would behoove the authors to discuss the results from Bechstedt and Brouhard 2012, which show that mutations in the C-DC disrupt cooperative interactions of DCX on the microtubule lattice. This study supports the authors’ current model greatly, yet it’s not discussed at all.

*Reviewer #2 (Recommendations for the authors):*

– Abstract: “The structural properties of MT-bound DCX remain poorly resolved.” Although a number of questions remain, this sentence does not, in my opinion, accurately reflect the current state of the field.

– Did the authors establish that isotope-labelled DCX is functionally equivalent to the “light” protein in the context of the assays shown in Figure S1?

– Gel lanes are mislabeled in Figure S1A.

– Figure S1E is provided to support the statement that no microtubule bundling is observed, but the resolution of the data are inadequate to support this statement – TIRF-MT data would be more suitable.

– Relatedly, these data are analysed to support the conclusion that increasing concentrations of DCX produce shorter MTs (Figure S1D) – what is the shortest polymer that was measured and how do this compare to what has been previously observed by TIRF-M?

– Figure S2 contains a lot of information visually but little is explained in the legend.

– What are the data depicted in Figure S3A+B a/d (and other similar panels in other figures)? 9 panels are typically shown, the legend for which is “The distribution of scores for the models computed for MT-NDC, using both crosslinking data and EM map (500,000 models).”

– Why are the panels in Figure S8Bd green/orange?

– “SI Methods” are mentioned on p27 – what does this refer to?

*Reviewer #3 (Recommendations for the authors):*

I am not an expert in microtubule biology, so I do not feel qualified to comment in detail about these aspects of the manuscript. However, I think that the conclusions are well balanced and the limitations of the model are adequately discussed.

---

## [Author Response]

Essential revisions:1) How accessible are the well-buried DCX-tubulin interfaces at the primary binding site to the chemical cross-linkers on which the analysis depends? As one reviewer points out, accessibility issues could explain the results depicted in Figure 3A, B, in which modelling that relies strictly on cross-links places NDC towards the outer edge of the protofilament, whereas inclusion of cryo-EM data in the integrated model places NDC in the inter-protofilament valley. The authors should discuss this further in the text and/or provide additional details about the measures they took to address accessibility issues.

There are no accessibility issues related to the crosslinks. In fact, we observe crosslinks to sites that are well buried in the cleft, as shown in Author response image 1. This is in line with data from a previous paper on MT crosslinking (Legal et al., 2016). The appearance of the models sitting near the outer edge of the protofilament is due to how we chose to represent the system, and is an expected edge effect (we added text to the top of page 10, indicating that it can be ignored). It is approximately half of the actual binding site and so expected to compete. To illustrate that accessibility is not an issue, we re-clustered the models with a lower threshold (2 Å) to generate smaller major cluster (22% of the total) where the NDC is positioned even more deeply within the inter-protofilament valley, as shown in Author response image 1. Clustering at higher threshold is preferred because it represents modelling uncertainty more faithfully by including the majority of the models generated during sampling.

**Author response image 1. sa2fig1:** (A) Crosslink sites on the MT lattice repeat unit highlighted in blue, showing that some are indeed buried within the interprotofilament groove. (B) Alternative representation showing the buried nature of NDC on the lattice.

2) It is important to repeat this analysis using the microtubule-bound structure of CDC (6RF2), given that this structure is conformationally distinct from PDB:5IP4. Based on analysis using the nanobody-bound CDC structure (5IP4), CDC appears to behave distinctly compared to NDC, such that CDC-derived cross-linking data are not consistent with the canonical inter-protofilament binding site. The authors should address if this depends on the particular PDB used.

We calculate the RMSD between 5IP4 and 6RF2 to be 5.1 Å, and show the alignment of the structures, see Author response image 2. This is a small difference when considering the precision of our integrative method, and thus would not change the results/conclusions presented in our paper. (Note that crosslinks are contrained with a distance of ~25 Å or less.) We have added a statement to the text to reflect this (page 10, bottom).

**Author response image 2. sa2fig2:** Structural alignment of the new MT-CDC structure (6RF2) to the one used in our study (5IP4), placed at the NDC binding site for illustration. CDC structures corresponding to 6RF2 and 5IP4 are shown with blue and cyan, respectively, α tubulins are shown in light grey and β tubulins are shown in dark grey, The RMSD calculated for residues 178-251 of the 5IP4 and 6RF2 is 5.1 Å.

3) Two reviewers noted that additional experiments would help support the observation and model that DCX oligomerises between the CDC domain and the C-terminal tail of the protein. The authors could either perform the crosslinking experiments with a previously published NDC-NDC chimera (Manka and Moores, 2020), which binds microtubules at the same inter-protofilament site but which lacks the CDC domain, or a DCX construct that lacks the C-terminal tail domain to examine if or how the interactions change.

This is an excellent idea. We elected to explore a disease-relevant truncation mutant, R303X (see des Portes V et al., 1998, Human Molecular Genetics, 7, 1063-70), which removes most of the tail and a large fraction of the crosslinked sites between the tails. We expressed both an mCherry-tagged wild type DCX and a GFP-tagged R303X mutant and explored the difference between the two in a flow-based TIRF experiment designed to highlight differences in cooperative binding to pre-formed microtubules. Interestingly, removal of the tail slightly reduces cooperative binding to the lattice. These observations indicate that the tail contributes to the self-assembly event and cooperativity, as the crosslinking data suggests. We have added considerable text to this effect in the manuscript on pp 16-18, provided a figure in the main text (new Figure 7), and added the appropriate methodological descriptions on pp 28, 30 and 31.

4) Certain technical details and quantification must be clarified or reported within the manuscript:The authors should indicate how many times were the crosslinking experiments repeated, and if they used more than one protein purification, which is ideal.

The crosslinking experiment was repeated several times during optimization. Then, the final results used in this work are the accumulated identifications across two technical replicates. The light-DCX protein used in this study was prepared in two separate batches, one to support optimization and one for final data analysis. For the final data, the heavy-DCX was also used from one batch, analyzed twice. Clarifying text is added to the manuscript on pages 29 and 32.

It is clear in the methods that the authors do not perform column chromatography to ensure homogeneity of the protein. It is important to provide evidence that the protein is well-behaved using size-exclusion chromatography, for example. SEC-MALS would be ideal to ensure there are not multiple oligomeric populations of DCX in their preps, but if the authors do not have access to this equipment, normal SEC is adequate.

If by homogeneity the reviewer means a lack of aggregation, we filtered our preps through spin cartridges but did not collect SEC data. Aggregates were not in evidence as shown by the lack of inter-DCX crosslinks in our prep. See Figure 2 —figure supplement 2 that extensively explores the possibility of aggregation explaining our XL-MS results. We are confident that none of our crosslinks reflect nonspecific aggregation of DCX.

The authors should provide quantification of S1C, especially if they are assuming a 1:1 stoichiometry. Please report how many times was this binding assay repeated.

The binding assay was repeated two times. Densitometry of the SDS-PAGE bands was performed and the results are shown in Figure 1 —figure supplement 1.

Please also report the average lengths +/- s.d. for S1D.

The average length for S1D is now in the legend:

“the average MT length was 5.0±3.3 and 2.6±1.7 µm, respectively”.

In addition, please provide zoom-in pictures of S1E for clear visualization of the microtubules.

We added zoomed-in views to Figure 1 —figure supplement 1.

Finally, do the authors think that varying salt concentration in their buffer during their crosslinking studies would affect oligomerisation of DCX? The authors should provide an explanation for why they chose to perform these studies in BRB80 and indicate whether they think varying the ionic strength would affect the inter-molecular interactions.

BRB80 was the buffer used in all assays performed in this manuscript – we kept it consistent throughout. We chose this buffer as it is the standard tubulin polymerization buffer and used in MT-DCX studies for many years (Bechstedt and Brouhard, 2012, Bechstedt et al., 2014, Manka and Moores, 2020). We note that BRB80 is not salt free: there is significant potassium as a result of titration to generate pH 6.8. (Sodium is usually excluded in biochemical preps of MTs.) As our assay data appears to faithfully represent the interaction work of others, we are comfortable with the consistency of the interaction that we have generated. It is possible that ionic strength can influence interactions but the wealth of MT-MAP studies using BRB80 gives us confidence in our results. For DCX-MT, note that Bechstedt *et al.,* have demonstrated that the preference of DCX for 13 pf is not influenced by ionic strength (Bechstedt and Brouhard, 2012).

5) The authors should discuss the results from Bechstedt and Brouhard 2012, which show that mutations in the C-DC disrupt cooperative interactions of DCX on the microtubule lattice. This study supports the authors' current model greatly, yet it's not discussed at all.

We did mention that these authors showed very strong cooperativity in wild-type binding, with Hill coefficients approaching 3 for DCX binding to MTs, but we did not mention the effects of mutations. We have updated the discussion to provide a more fulsome treatment, building from our new observations that the R303X mutant slightly weakens the interaction on the MT lattice. The Bechstedt study explored 7 mutations across the CDC (but none in the tails), which also demonstrated reduced cooperativity in binding. Taken together, the evidence shows that both the CDC and the tail play a role in maintaining a high avidity interaction. We have added text to this effect on pp 19,20.

Reviewer #1 (Recommendations for the authors):1) The authors should provide quantification of S1C, especially if they are assuming a 1:1 stoichiometry. How many times was this binding assay repeated? The average lengths are not reported in the manuscript for S1D. Provide zoom-in pictures of S1E, since it is impossible to see the microtubules in this image.

The binding assay was repeated two times. Densitometry of the SDS-PAGE bands was performed and the results are shown in Figure 1 —figure supplement 1. The average lengths are now reported in the legend and zoomed-in views are also provided.

2) How many times were the crosslinking experiments repeated? Did the authors use more than one protein purification?

The crosslinking experiment was repeated several times during optimization. Then, the final results used in this work are the accumulated identifications across two technical replicates. The light-DCX protein used in this study was prepared in two separate batches, one to support optimizaton and one for final data analysis. For the final data, the heavy-DCX was also used from one batch, analyzed twice. Clarifiying text is added to the manuscript.

3) In Figure S6, the authors show that only under extreme conditions, such as denaturation, can DCX self-associate in the absence of tubulin. It appears as if the authors performed their crosslinking studies in BRB80 without additional salt. Are the authors able to perform the crosslinking studies at a more physiological salt concentration? Or could the authors provide an explanation as to why they do not think this would affect the inter-molecular interactions?

BRB80 was the buffer used in all assays performed in this manuscript – we kept it consistent throughout. We chose this buffer as it is the standard tubulin polymerization buffer and used in MT-DCX studies for many years (Bechstedt and Brouhard, 2012, Bechstedt et al., 2014, Manka and Moores, 2020). We note that BRB80 is not salt free: there is significant potassium as a result of titration to generate pH 6.8. (Sodium is usually excluded in biochemical preps of MTs.) As our assay data appears to faithfully represent the interaction work of others, we are comfortable with the consistency of the interaction that we have generated. It is possible that ionic strength can influence interactions but the wealth of MT-MAP studies using BRB80 gives us confidence in our results. For DCX-MT, note that Bechstedt *et al.,* have demonstrated that the preference of DCX for 13 pf is not influenced by ionic strength (Bechstedt and Brouhard, 2012).

4) It would be useful to test the model generated in this study by performing certain experiments with a DCX construct that lacks the C-terminal tail. It is obviously difficult to evaluate microtubule binding without either DC domain, but if a tail-less DCX is still able to bind to microtubules, then it could be very useful in providing support for this model. Would a tail-less DCX exhibit less cooperative binding? Would removing the tail abolish the inter-molecular interactions? Or would the C-DC domains still be able to self-associate?

This is an excellent idea. We elected to explore a disease-relevant truncation mutant, R303X (see des Portes V et al., 1998, Human Molecular Genetics, 7, 1063-70), which removes most of the tail and a large fraction of the crosslinked sites between the tails. We expressed both an mCherry-tagged wild type DCX and a GFP-tagged R303X mutant and explored the difference between the two in a flow-based TIRF experiment designed to highlight differences in cooperative binding to pre-formed microtubules. Interestingly, removal of the tail slightly reduces cooperative binding to the lattice. These observations indicate that the tail contributes to the self-assembly event and cooperativity, as the crosslinking data suggests. We have added considerable text to this effect in the manuscript, provided a figure in the main text (new Figure 7), and added the appropriate methodological descriptions.

5) It would behoove the authors to discuss the results from Bechstedt and Brouhard 2012, which show that mutations in the C-DC disrupt cooperative interactions of DCX on the microtubule lattice. This study supports the authors' current model greatly, yet it's not discussed at all.

We did mention that these authors showed very strong cooperativity in wild-type binding, with Hill coefficients approaching 3 for DCX binding to MTs, but we did not mention the effects of mutations. We have updated the discussion to provide a more fulsome treatment, building from our new observations that the R303X mutant slightly weakens the interaction on the MT lattice. The Bechstedt study explored 7 mutations across the CDC (but none in the tails), which also demonstrated reduced cooperativity in binding. Taken together, the evidence shows that both the CDC and the tail play a role in maintaining a high avidity interaction. We have added text to this effect.

Reviewer #2 (Recommendations for the authors):– Abstract: "The structural properties of MT-bound DCX remain poorly resolved." Although a number of questions remain, this sentence does not, in my opinion, accurately reflect the current state of the field.

We appreciate the progress made by many research groups in this area. We simply meant there are still unanswered questions of a structural nature. The text is modified to indicate that the structural properties of MT-bound DCX that explain the clinical disorders are incompletely determined.

– Did the authors establish that isotope-labelled DCX is functionally equivalent to the "light" protein in the context of the assays shown in Figure S1?

No, we did not, but isotope labeling (changing ^14^N to ^15^N) does not change the physicochemical properties of proteins.

– Gel lanes are mislabeled in Figure S1A.

Thanks for noting this. The correction is made to the figure.

– Figure S1E is provided to support the statement that no microtubule bundling is observed, but the resolution of the data are inadequate to support this statement – TIRF-MT data would be more suitable.

We provide a zoomed-in view in Figure 1 —figure supplement 1 to support our observation that extensive bundling is absent (and we added the word “extensive”).

– Relatedly, these data are analysed to support the conclusion that increasing concentrations of DCX produce shorter MTs (Figure S1D) – what is the shortest polymer that was measured and how do this compare to what has been previously observed by TIRF-M?

The shortest MT length that we analyzed was 0.7 µm, but we note that growth conditions are very difficult to compare across studies, as many factors control length. We simply note this shortening in a relative sense.

– Figure S2 contains a lot of information visually but little is explained in the legend.

We have added additional text to the figure caption to better clarify the four steps in a typical integrative structural modeling exercise. We refer the reader to the extensive methods description that accompanies these figures.

– What are the data depicted in Figure S3A+B a/d (and other similar panels in other figures)? 9 panels are typically shown, the legend for which is "The distribution of scores for the models computed for MT-NDC, using both crosslinking data and EM map (500,000 models)."

These panels are the statistical output generated by IMP for each modeling scenario. We present these data to support the fact that sufficient sampling was performed in the integrative modeling, and convergence was obtained. The reviewer is referring to the meaning of the individual plots. The supplemental methods section describes the process of generating these figures in greater depth and provides references to these complex metrics. It is more than we can reasonably include in the captions.

– Why are the panels in Figure S8Bd green/orange?

The first few steps of the clustering analysis is usually performed in multiple rounds to ensure sampling convergence is achieved. Although each round may produce a different color scheme, the different colors don’t mean anything specific as they all represent all of the data for each score.

– "SI Methods" are mentioned on p27 – what does this refer to?

Sorry, that was a misplaced reference to the methodology. It has been removed.

References:

Bechstedt, S. & Brouhard, G. J. 2012. Doublecortin recognizes the 13-protofilament microtubule cooperatively and tracks microtubule ends. Developmental cell, 23, 181-192.

Bechstedt, S., Lu, K. & Brouhard, G. J. 2014. Doublecortin recognizes the longitudinal curvature of the microtubule end and lattice. Current Biology, 24, 2366-2375.

Legal, T., Zou, J., Sochaj, A., Rappsilber, J. & Welburn, J. P. I. 2016. Molecular architecture of the Dam1 complex–microtubule interaction. Open Biology, 6, 150237.

Manka, S. W. & Moores, C. A. 2020. Pseudo‐repeats in doublecortin make distinct mechanistic contributions to microtubule regulation. EMBO reports, 21, e51534.